# PROBING THE BOUNDARIES OF SCIENTIFIC CONCEPTS IN LANGUAGE MODELS

## ABSTRACT

Systematic investigation of the understanding of scientific concepts has not received much attention in language models. This gap can be bridged by a formalized theory of conceptual semantics that maps naturally to instruction templates for natural language agents. We propose a simple framework expressible in first-order logic to address the semantic compositionality of scientific concepts, noun phrases and conceptual hierarchies. The framework is used to derive a conceptual integrity benchmark with 6 tasks that are applied to a selection of 187 concepts from the domains of biology, chemistry and medicine. The performance of 15 state-of-the art language models is evaluated relative to baseline information collected from various knowledge repositories. We see a strong positive correlation between model size and performance. External validity of the benchmark is demonstrated by a high correlation with other benchmarks that measure related skills. It is suggested that the proposed framework and associated benchmark provide a practical template for developing conceptual integrity benchmarks in a wide array of technical or scientific domains.

## 1 INTRODUCTION

Largest of the available language models (LMs) have been trained in a self-supervised manner on a significant part of the publicly available text on the internet. They are able to absorb large amounts of information which turns instruction-tuned language models into broad-purpose knowledge bases and reasoners that can be queried using natural language. In order to support research, LMs are required to integrate information about domain-relevant concepts in terms of definitions, examples/instances and relationships to related concepts. Such information is scattered across many documents and there is no *a priori* justification that a language model is able to pull it all together into a complete model of a concept. Here, we propose the term **conceptual integrity** to cover the set of reciprocal associations between a concept label, its definition and its referents (Figure 1). Before using LMs as knowledge bases or reasoning agents, it is crucial to benchmark them for conceptual integrity in the domains of interest since the validity of inference strictly depends on the accuracy of the reasoner's grasp of the concepts involved.

We focus specifically on concepts with well-defined boundaries, expert-curated definitions, and enumerable referents. This contrasts with everyday concepts which exhibit graded category membership, prototype effects, and lack authoritative consensus annotations. While our theoretical framework is domain-agnostic, the present application targets a selection of scientific domains relevant to pharmaceutical R&D (biology, chemistry, and medicine) where ground truth can be reliably established based on expert-curated consensus annotations.

Benchmarking LMs on various aspects of language understanding (e.g., reading comprehension Paperno et al. (2016); Dua et al. (2019), common knowledge and reasoning Sakaguchi et al. (2021); Wei et al. (2024), problem solving Hendrycks et al. (2021a), abstract reasoning Liu et al. (2021); Shi et al. (2023), specialized knowledge and reasoning Rein et al. (2024); Auer et al. (2023), etc.) has received considerable attention while systematic investigation of conceptual understanding has remained largely out of focus. We suspect it is due to the lack of a comprehensive framework of conceptual semantics to suggest measures of conceptual integrity. Well-known models of semantics have been successful in capturing isolated aspects of semantic compositionality Szabó (2024) on concept (distributional semantics Mikolov et al. (2013); Boleda (2020)), conceptual hierarchy (description

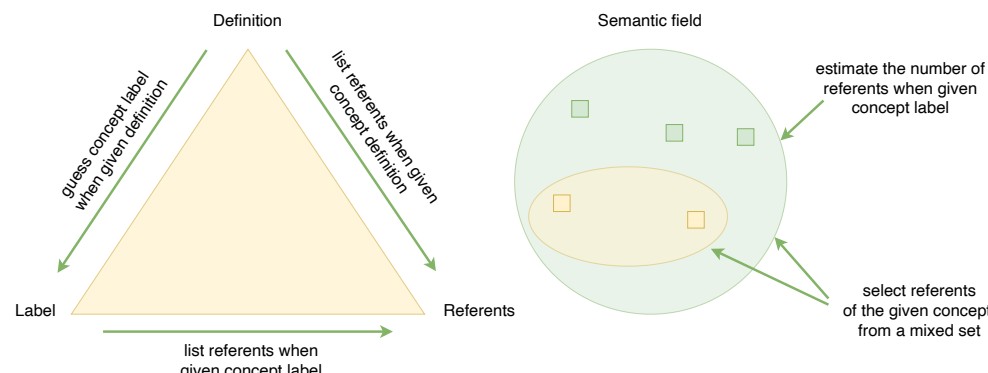

Figure 1: **Tests of conceptual integrity.** Conceptual integrity requires coherent associations between the *concept label* (linguistic term), *definition* (selection criteria), and *referents* (instances satisfying the criteria).

logics/ontologies Baader et al. (2004)) and proposition levels (Montague semantics Montague (1970); Janssen (2014)) while not offering a straightforward path to the unification of these aspects Boleda & Herbelot (2016). At least presently, we are not aware of a formalized theory of conceptual semantics that can be easily mapped to instruction templates covering essential aspects of conceptual integrity in natural language agents (e.g., humans and LMs).

We suggest that a formalization of conceptual semantics applicable to conceptual integrity testing should exhibit the following properties:

1. Mappings between symbol, concept and referents of the concept
2. Semantic compositionality on concept level (e.g., definitions of concepts)
3. Semantic compositionality on phrase level (e.g., noun and verb phrases)
4. Support of conceptual hierarchies (e.g., nested classes, ontologies)
5. Semantic compositionality on proposition level (e.g., subject, object and predicate)

Here, we outline and apply a simple framework expressible in first-order logic that addresses requirements 1–4 with the compositionality of verb phrases and property 5 left as aspirational goals for future development.

Both informal and formal expositions of the framework are provided. Consistency of the proposed model of semantic compositionality on concept and noun phrase levels is demonstrated via provable properties as Lean 4 code Moura & Ullrich (2021). To illustrate its applicability, the framework is used to generate 6 related tasks which apply to any concept with a definition and enumerable referents. We benchmark the conceptual integrity of 15 state-of-the-art LMs of different sizes (both open and closed weight) on a selection of 187 concepts relevant to the pharmaceutical industry from the domains of chemistry, biology and medicine. We use definitions and canonical lists of referents from authoritative sources such as Wikipedia, research databases, human-curated ontologies and research papers as a baseline for scoring performance. We demonstrate that the conceptual integrity exhibited by a model correlates well with its ranking in natural language understanding benchmarks that employ similar tasks while correlating less with benchmarks based on remotely related tasks.

## 2 FRAMEWORK OF CONCEPTUAL SEMANTICS

We distinguish three aspects of linguistic reference: **concept label** (linguistic term, e.g., "cytokine"), **definition** (selection criteria), and **referents** (entities satisfying the criteria, e.g., TNF-$\alpha$, IL-6). These form a reference chain $S \implies C \implies R$ following the tradition of formal semantics (Frege, 1892; Carnap, 1947). Colloquial usage might conflate these distinctions but we consider them necessary for rigorous semantic analysis.

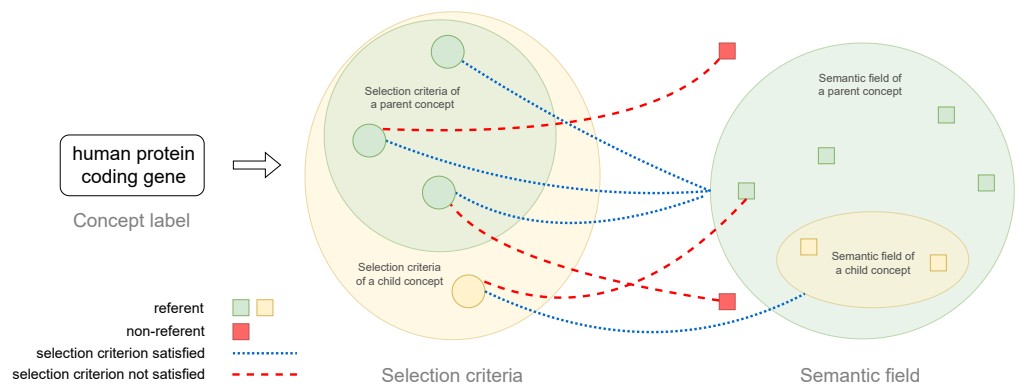

Figure 2: **Logical framework of conceptual semantics.** The concept label implies a set of selection criteria that implies the semantic field. The association between concept label and selection criteria is arbitrary (languages use different terms for the same concept). Selection criteria restrict the semantic field of the concept to entities that satisfy the selection criteria.

A referent is an entity to which a linguistic expression refers. The proposed logical framework of conceptual semantics defines a concept as a set of **selection criteria** for its referents (Figure 2). Selection criteria are propositions that must be jointly satisfied by the referents. Thus, selection criteria constitute the definition of the concept and they restrict its **semantic field** (the set of entities that the concept refers to). The chain of linguistic reference (linguistic implication) is defined as $S \implies C \implies R$ where $S$ is symbol (e.g., a linguistic term), $C$ is concept (e.g., a definition) and $R$ is the semantic field of concept $C$ (i.e. entities satisfying the definition).Full formalism (precise definitions of concepts and semantic fields, identity and equivalence on selection criteria and concepts), together with proofs of properties (e.g. the compositionality of concepts and meaning), is provided in Supplementary Section A.1.

## 3 DATASET AND BENCHMARKING

### 3.1 CONCEPTS

In total 187 concepts were selected from the domains of chemistry, biology and medicine spanning 3 to 151,204 referents per concept (Supplementary Table 7). These domains were selected beyond practical relevance to pharmaceutical R&D: (1) availability of human expert-curated ground truth annotations for concepts (e.g., from ChEBI, MeSH, Gene Ontology), (2) well-defined referents with discrete category membership and consistency between definitions and class members, (3) conceptual diversity across scales (molecular to biological to clinical), and (4) transferability to several key scientific domains (chemistry, biology, medicine).

Each concept was annotated in terms of definition, selection criteria (as derived from the definition) and referents (see section B.1 in Supplementary for more information). The selection of concepts included in the dataset is largely arbitrary and by no means exhaustive or comprehensive in terms of importance to drug discovery. The number of concepts subjected to each task is detailed in Supplementary Table 8.

### 3.2 TESTS

Based on the proposed logical framework of conceptual semantics, we formulated six related tests as instruction templates to the LM. Together, these tests cover the critical relationships between the concept's label, definition and referents that govern conceptual integrity (Figure 1).

The first two tests ask the test subject to name the concept either based on its definition from a dictionary, ontology or similar source ('decide-concept', B.2.1) or based on a set of selection criteria

for the referents of the concept ('decide-concept-from-selection-criteria', B.2.2). These tests evaluate the compatibility of dictionary definitions with explicit selection criteria.

The second pair of tests asks the test subject to provide up to $n$ instances (referents) of the concept based on the concept label ('limited-list-referents', B.2.3) or selection criteria ('limited-list-referents-from-selection-criteria', B.2.4). Results from these tests assess whether concept labels and selection criteria exhibit compatibility in eliciting appropriate referents. We opted for limited enumeration of referents (with $n = 24$), because exhaustive enumeration is unwieldy for concepts with large semantic fields and tended to induce looping (repetition) behavior in some models while also being potentially very costly in terms of the number of output tokens produced.

The 'decide-referents' (B.2.5) task tests the ability of the model to identify examples of the concept from a selection of alternatives from sibling concepts (conceptual categories lying at the root of the same parent concept). In total, 24 referents per prompt were sampled in equal proportions from 3-6 sibling concepts depending on their availability in the parent concept. Due to a lack of nested structure of referents or a limited number of sibling concepts with sufficient number of children, this test was applicable to a subset of concepts with compatible referent trees. In most cases, these concepts came from the domain of chemistry. The task prompt contextualized with a sampled subset of referents from $k$ sibling concepts was used to generate $k$ different prompts, each asking to identify referents of the particular subconcept $i \in k$ from the given list.

Finally, the 'semantic-field-size' (B.2.6) test asks to estimate the number of referents for a concept by providing a point estimate (a specific number) and a range in orders of magnitude up to a billion or more (see the prompt template for details).

## 3.3 BENCHMARKING

We benchmarked 15 open and closed source LMs of various sizes (Table 1). In total, the models were subjected to 14,490 tests (966 tests per model) covering 187 concepts. Scoring was performed using the LLM-as-a-judge approach as detailed in Supplementary section B.3. Average and domain-specific model performance was obtained on a subset of responses that did not include the 'semantic-field-size' task because it was found to be weakly correlated with other tasks for reasons that will be outlined below.

Model performance across the studied domains has been summarized in Table 1 and Supplementary Figure 4. In general, larger variants of the studied closed source models exhibited top performance. Performance of models across tasks has been summarized in Figure 3. Kendall tau rank correlation (interpreted based on the cutoffs from Wicklin (2023)) between domain competence in biology, chemistry and medicine bordered on moderate/strong (0.67, upper left quadrant of Table 2). This suggests that better performing models tend to excel across the board on the domains of interest.

In contrast, correlations between external benchmarks (lower right quadrant of Table 2) ranged from weak to strong (0.31-0.70), suggesting that external benchmarks target a range of partially overlapping skill sets. The average performance on the conceptual integrity benchmark was very strongly correlated with MMLU (0.82), moderately correlated with GPQA (0.39) and DROP (0.48) and weakly correlated with the MATH (0.28) benchmark. The strength of these correlations aligns well with the extent of overlap between skill sets assessed in the current and external benchmarks. MMLU Hendrycks et al. (2021a) evaluates mostly factual knowledge in terms of multiple choice questions from the domains of humanities, social and natural sciences. GPQA Rein et al. (2024) contains challenging questions from biology, physics, and chemistry that require specialized reasoning skills while DROP Dua et al. (2019) is a set of general reading comprehension and basic reasoning tasks. MATH Hendrycks et al. (2021b) is a set of challenging competition mathematics problems requiring mathematical reasoning that is most remote from the current benchmark which evaluates basic understanding of biomedical and chemistry concepts.

Internal consistency of the benchmark was assessed based on performance correlation between tasks (Table 3). Very high correlation (0.87) between model ranks from the two 'decide-concepts' tasks indicates compatibility between definitions and corresponding selection criteria when naming concepts. In general, model performance ranking on all tasks except for the 'semantic-field-size' exhibited strong to very strong correlation (0.61-0.87). It appears to suggest that these tasks are indeed measuring related competences although correlations do not imply a causal interpretation.

Table 1: **Overview of benchmarked language models.** The table includes size, open weight status (OW), and task-specific performance across domains relevant to pharmaceutical R&D. "Avg" indicates average performance across the studied domains (Biology, Chemistry, Medicine), while "SFS" refers to accuracy on the 'semantic-field-size' task. Model sizes are abbreviated as S (small), M (medium), and L (large). OW is marked with ✓ for open weight models and ✗ for closed weight models.

| Model | Size | OW | Avg | Bio | Chem | Med | SFS |
|---|---|---|---|---|---|---|---|
| claude-3-5-sonnet | M [3] | ✗ | 0.698 | 0.725 | 0.643 | 0.726 | 0.533 |
| claude-3-haiku | S [3] | ✗ | 0.641 | 0.705 | 0.564 | 0.654 | 0.627 |
| claude-3-opus | L [3] | ✗ | 0.695 | 0.742 | 0.647 | 0.697 | 0.480 |
| claude-3-sonnet | M [3] | ✗ | 0.658 | 0.697 | 0.581 | 0.695 | 0.552 |
| gemma-3 | M (27B) | ✓ | 0.653 | 0.725 | 0.579 | 0.656 | 0.493 |
| gpt-35-turbo | M [3] | ✗ | 0.604 | 0.690 | 0.503 | 0.619 | 0.418 |
| gpt-4 | L [3] | ✗ | 0.670 | 0.732 | 0.583 | 0.696 | 0.562 |
| gpt-4o | L [3] | ✗ | 0.702 | 0.746 | 0.636 | 0.725 | 0.475 |
| gpt-4o-mini | S [3] | ✗ | 0.644 | 0.717 | 0.535 | 0.680 | 0.632 |
| llama3-70b-instruct | M (70B) | ✓ | 0.639 | 0.697 | 0.561 | 0.661 | 0.480 |
| llama3-8b-instruct | S (8B) | ✓ | 0.533 | 0.601 | 0.419 | 0.578 | 0.457 |
| mistral-small-instruct-24B | M (24B) | ✓ | 0.669 | 0.732 | 0.611 | 0.665 | 0.479 |
| o1-mini | L [3] | ✗ | 0.688 | 0.732 | 0.630 | 0.703 | 0.476 |
| phi-v4 | S (14B) | ✓ | 0.633 | 0.680 | 0.540 | 0.679 | 0.492 |
| qwen-v2.5-14b-instruct | S (14B) | ✓ | 0.603 | 0.692 | 0.484 | 0.632 | 0.447 |

Table 2: **Correlation of internal and external performance rankings** (Avg: Average, Bio: Biology, Chem: Chemistry, Med: Medicine, HumEv: HumanEval).

| | Avg | Bio | Chem | Med | MMLU | GPQA | MATH | HumEv | DROP |
|---|---|---|---|---|---|---|---|---|---|
| Avg | 1.00 | 0.83 | 0.72 | 0.83 | 0.82 | 0.39 | 0.28 | 0.67 | 0.48 |
| Bio | 0.83 | 1.00 | 0.67 | 0.67 | 0.65 | 0.33 | 0.22 | 0.61 | 0.42 |
| Chem | 0.72 | 0.67 | 1.00 | 0.67 | 0.82 | 0.44 | 0.11 | 0.50 | 0.37 |
| Med | 0.83 | 0.67 | 0.67 | 1.00 | 0.87 | 0.56 | 0.33 | 0.83 | 0.54 |
| MMLU | 0.82 | 0.65 | 0.82 | 0.87 | 1.00 | 0.54 | 0.31 | 0.70 | 0.43 |
| GPQA | 0.39 | 0.33 | 0.44 | 0.56 | 0.54 | 1.00 | 0.67 | 0.61 | 0.42 |
| MATH | 0.28 | 0.22 | 0.11 | 0.33 | 0.31 | 0.67 | 1.00 | 0.50 | 0.31 |
| HumEv | 0.67 | 0.61 | 0.50 | 0.83 | 0.70 | 0.61 | 0.50 | 1.00 | 0.48 |
| DROP | 0.48 | 0.42 | 0.37 | 0.54 | 0.43 | 0.42 | 0.31 | 0.48 | 1.00 |

Performance ranking in the 'semantic-field-size' task (Table 1) exhibited negligible to weak correlation (0.03 to 0.26) with other tasks. To investigate whether model failure, gold standard incompleteness or instruction ambiguity explains these anomalies, we conducted a detailed analysis comparing normalized discrepancies between point estimates and gold standard semantic field sizes depending on the magnitude of concept's semantic field (Supplementary Section B.7). This analysis revealed that a major issue was task instruction ambiguity arising from fundamental differences in ontological structure across domains, particularly for biological cell type concepts where models confused class enumeration with physical instance counting. For example, when asked about "lymphocyte" semantic field size, models often responded with estimates in the billions or trillions (reflecting the total number of lymphocytes in the human body) rather than 3-5 (the number of major lymphocyte subtypes listed in the gold standard).

To investigate whether various characteristics of concepts influence task performance, a statistical analysis of category-level effects across five categorization dimensions (abstraction level, semantic field size, domain, ontological structure, and number of selection criteria) was conducted. Overall, abstraction level and semantic field size had the largest and most consistent effects across tasks (detailed analysis in Appendix B.6).

---

[3]Tentative size estimate (information not public).

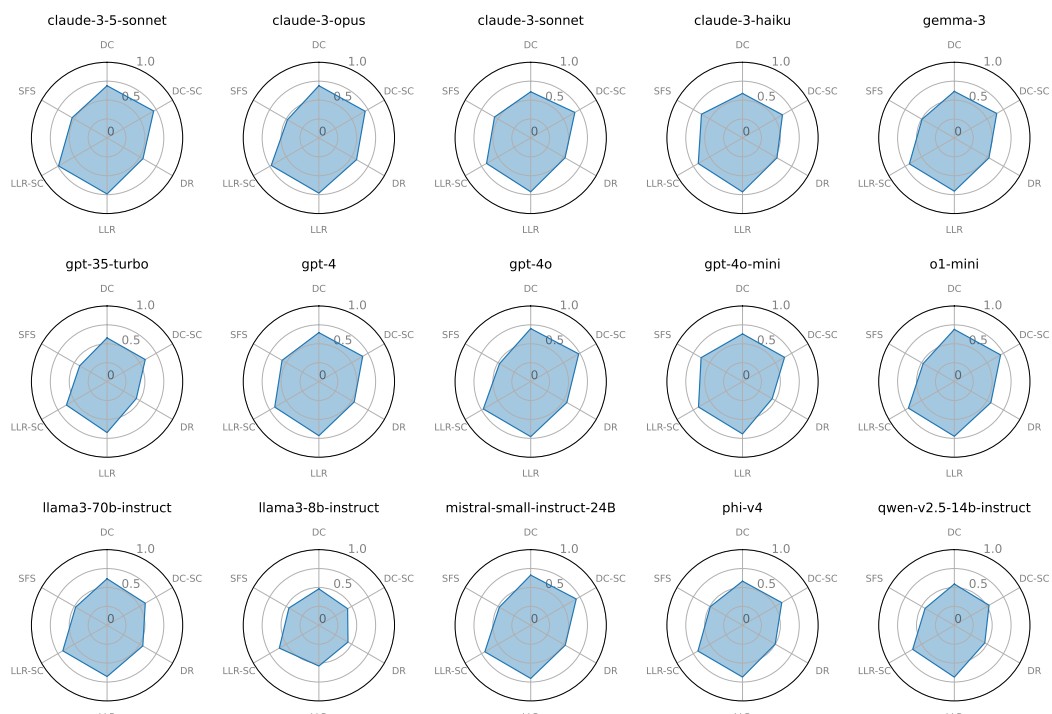

Figure 3: **Performance of models on tests of conceptual integrity.** Each axis represents accuracy on a different task that evaluates conceptual integrity. Acronyms for tasks: DC: decide-concept, DC-SC: decide-concept-from-selection-criteria, DR: decide-referents, LLR: limited-list-referents, LLR-SC: limited-list-referents-from-selection-criteria, SFS: semantic-field-size. See Supplementary Methods for details on scoring.

Table 3: Correlation between models' performance on different tasks. **DC**: decide-concept, **DC-SC**: decide-concept-from-selection-criteria, **LLR**: limited-list-referents, **LLR-SC**: limited-list-referents-from-selection-criteria, **DR**: decide-referents, **SFS**: semantic-field-size

| oprule | DC | DC-SC | LLR | LLR-SC | DR | SFS |
|---|---|---|---|---|---|---|
| DC | 1.00 | 0.87 | 0.66 | 0.64 | 0.68 | 0.13 |
| DC-SC | 0.87 | 1.00 | 0.75 | 0.71 | 0.61 | 0.13 |
| LLR | 0.66 | 0.75 | 1.00 | 0.66 | 0.64 | 0.28 |
| LLR-SC | 0.64 | 0.71 | 0.66 | 1.00 | 0.68 | 0.05 |
| DR | 0.68 | 0.61 | 0.64 | 0.68 | 1.00 | -0.01 |
| SFS | 0.13 | 0.13 | 0.28 | 0.05 | -0.01 | 1.00 |

## 3.4 IDENTIFICATION OF HIGH AND LOW PERFORMERS

Inspired by a method for nonparametric estimation of differential expression used in Ilmjärv et al. (2014), we resorted to the Fisher's hypergeometric test Beal (1976) to highlight models that stood out from the group in terms of their performance. Our analysis indicates that the average performance of Gpt-4o, Claude 3.5 Sonnet, Claude 3 Opus in the benchmark is virtually identical while o1-mini also stands out at the top (Table 4). On the other end, Gpt-3.5-turbo, Qwen-v2.5-14b-instruct and especially Llama3-8b-instruct stood out as poor performers. For more information regarding the rationale behind the applicability of the hypergeometric test, please see Supplementary Methods B.5.

Table 4: Statistical outliers in group performance based on the hypergeometric test. P-values were corrected for multiple testing using the FDR-procedure of Benjamini-Yekuteli. Outliers and significant q-values are in bold face.

| | correct | correct (%) | p (upper) | q (upper) | p (lower) | q (lower) |
|---|---|---|---|---|---|---|
| **gpt-4o** | 659 | 68.2 | 6.06E-05 | **1.34E-03** | 1.00 | 1.00 |
| **claude-3-5-sonnet** | 658 | 68.1 | 8.05E-05 | **1.34E-03** | 1.00 | 1.00 |
| **claude-3-opus** | 658 | 68.1 | 8.05E-05 | **1.34E-03** | 1.00 | 1.00 |
| **o1-mini** | 648 | 67.1 | 1.07E-03 | **1.33E-02** | 9.99E-01 | 1.00 |
| mistral-small-instruct-24B | 631 | 65.3 | 2.97E-02 | 2.96E-01 | 9.70E-01 | 1.00 |
| gpt-4 | 623 | 64.5 | 9.18E-02 | 7.61E-01 | 9.08E-01 | 1.00 |
| claude-3-sonnet | 613 | 63.5 | 2.62E-01 | 1.00 | 7.38E-01 | 1.00 |
| gemma-3 | 611 | 63.3 | 3.08E-01 | 1.00 | 6.92E-01 | 1.00 |
| claude-3-haiku | 598 | 61.9 | 6.53E-01 | 1.00 | 3.47E-01 | 1.00 |
| llama3-70b-instruct | 595 | 61.6 | 7.26E-01 | 1.00 | 2.74E-01 | 1.00 |
| gpt-4o-mini | 591 | 61.2 | 8.09E-01 | 1.00 | 1.91E-01 | 1.00 |
| phi-v4 | 584 | 60.5 | 9.12E-01 | 1.00 | 8.79E-02 | 1.00 |
| **gpt-35-turbo** | 556 | 57.6 | 9.99E-01 | 1.00 | 5.56E-04 | **9.22E-03** |
| **qwen-v2.5-14b-instruct** | 549 | 56.8 | 1.00 | 1.00 | 9.37E-05 | **2.33E-03** |
| **llama3-8b-instruct** | 482 | 49.9 | 1.00 | 1.00 | 7.93E-17 | **3.95E-15** |

## 3.5 IDENTIFICATION OF FAILURE MODES

### 3.5.1 CONCEPT ANNOTATION FAILURE

The following example is based on the annotation of the concept *antioxidant system protein* to demonstrate that care needs to be taken to align the concept and its definition to the required degree of precision.

When an LM was asked to provide the concept label based on the definition or selection criteria (Appendix C.1), the response was predominantly "antioxidant protein" which is accurate given the definition "A protein with antioxidant activity". However, the LM judge (Gpt-4o) labeled such responses as incorrect with respect to the ground truth of "antioxidant system protein" and provided the following reason:

> "An antioxidant protein is a type of protein that prevents oxidation, but not all antioxidant system proteins are exclusively antioxidant proteins. The antioxidant system may include various components, not limited to individual antioxidant proteins."

Since transcription factors (e.g., NRF2) that regulate the abundance of antioxidant proteins belong to the antioxidant system while not being proteins with antioxidant activity themselves, it is indeed justified to regard the concepts *antioxidant protein* and *antioxidant system protein* as not strictly equivalent. On the other hand, it is rather likely that researchers who are not experts of antioxidant biology would tend to overlook this semantic nuance and consider the terms "antioxidant system protein" and "antioxidant protein" equivalent. This suggests it can be helpful to consult LMs when annotating concepts to make sure that the concept label and definition are aligned and to require brief justifications from scoring models to aid traceability.

### 3.5.2 TASK INSTRUCTION AMBIGUITY ACROSS DOMAINS

The semantic field size (SFS) task revealed a systematic failure mode arising from ambiguous instructions when applied across domains with different ontological conventions. The task prompt instructed models to "estimate the size of the semantic field for a given concept" and specified: "If the concept is a class consisting of subclasses, report the number of subclasses." For chemical concepts where ontologies enumerate distinct molecular structures, this instruction yielded consistent results. However, for biological cell type concepts, the instruction proved fundamentally ambiguous.

For example, the MeSH taxonomy lists 3 major functionally distinct subtypes of "lymphocyte" (see Appendix C.3 for the detailed example). However, models frequently interpreted the question as asking for the number of individual lymphocyte cells in the human body—a valid alternative interpretation given that lymphocytes are physically countable entities with approximately $2 \times 10^{12}$ instances *in vivo* Alberts et al. (2002).

This ambiguity is not a model failure per se, but rather a limitation of the task design when applied to conceptual hierarchies with multiple levels of abstraction. Chemical structures in ChEBI are enumerated as distinct referents (e.g., D-ribose 5-phosphate vs. D-ribose 1-phosphate are separate entities), whereas biological cell types in GO and MeSH are organized taxonomically with "children" representing subclasses rather than individual instances. The systematic nature of this failure mode across various model sizes highlights that domain-specific differences in the nature of the ontological information represented by the gold standard must be taken into account when drafting instruction templates for the task in question.

### 3.5.3 RESPONSE SCORING FAILURE

The following example illustrates that overly stringent instructions for the assessment of conceptual equivalence can conflict with standard term usage. When LMs were asked to provide a concept label corresponding to the definition "structural or functional unit of the brain" they often responded with "brain region". The LM judge considered the response "brain region" as not equivalent to "brain structure" in the context of the evaluation instructions because

> "A brain region is a specific area within the brain, whereas a brain structure can refer to any anatomical feature within the brain, including but not limited to regions. Therefore, not all brain structures are brain regions".

Essentially, the judge argued that a brain region is a location within the brain (e.g., medial preoptic area) whereas a brain structure (e.g., medial forebrain bundle) can span multiple regions. While this is semantically correct, the terms "brain region" and "brain structure" are typically used interchangeably by anatomists and neuroscientists to whom the pursuit of semantic rigor is not absolutely critical.

Similarly, the LM judge did not consider concept labels "lung cancer medication" and "lung cancer drug" to be equivalent to the baseline label "approved drug for lung cancer" because

> "A lung cancer drug may not necessarily be approved, while an approved drug for lung cancer is specifically one that has received approval for treating lung cancer".

These examples suggest that scoring instructions to the LM judge could be rendered more practical by asking the judge to assess the equivalence of concepts from the viewpoint of a domain expert or practitioner ("someone skilled in the art") instead of strictly focusing on semantic rigor.

### 3.5.4 LANGUAGE MODEL FAILURE

A consistent failure mode was identified in smaller versions of state-of-the-art models (Gpt-4o vs Gpt-4o-mini, Claude-3 Sonnet vs Haiku, Llama 3 70B vs 8B) when listing referents of "cyclic nucleotide" based on the following selection criteria:

> "it is a single-phosphate nucleotide",
> "it has a cyclic bond arrangement between the sugar and phosphate groups"

While both larger and smaller versions listed predominantly cyclic nucleotides when prompted with the concept label "cyclic nucleotide", smaller models started the response with non-cyclic variants of nucleotides when prompted with the selection criteria quoted above (see Appendix C.2). Since selection criteria are an unconventional representation of definitions, it seems plausible that smaller models were unable to fully condition the probability of referents based on the second selection criterion despite the task instructions insisting that

> "The definition is given as a list of necessary and sufficient criteria that each referent must satisfy".

## 4 DISCUSSION AND RELATED WORK

We have proposed a logical framework of the compositional semantics of scientific concepts that includes concept labels, definitions and referents. Based on this framework, a conceptual integrity benchmark was designed and applied to measure the degree of correspondence between LM responses and baseline annotations from sources representing expert-curated consensus (e.g. databases, ontologies and research papers). The external validity of the benchmark is demonstrated by the consistent degradation of conceptual integrity measure in smaller instances of the same model. Thus, the ordering of benchmark scores of related instances of Claude 3 (Opus > Sonnet > Haiku), Gpt (4o > 4o mini) and Llama 3 (70B > 8B) were consistent with model size.

The conceptual integrity framework complements prior work on evaluating conceptual knowledge in language models, particularly COPEN Peng et al. (2022) and OntoProbe Wu et al. (2023). While these frameworks share the goal of assessing conceptual understanding, they address distinct aspects: COPEN evaluates taxonomic similarity and property knowledge using multiple-choice classification tasks on general concepts from Wikipedia/DBpedia; OntoProbe assesses ontological structure and logical reasoning through cloze-completion tasks testing RDFS entailment rules. In contrast, our framework focuses specifically on definitional coherence in the form of bidirectional mappings between concept labels, selection criteria, and referents enabling systematic derivation of evaluation tasks from formal principles. Conceptual integrity benchmark differentiates itself by including generative enumeration of referents from concept labels or selection criteria, assessing productive knowledge rather than only recognition or classification. In addition, it leverages expert-curated scientific databases representing a consensus to provide high-confidence ground truth for technical concepts with well-defined boundaries.

We suggest that conceptual integrity is a foundational capability that is both distinguishable from and prerequisite to logical and factual reasoning. The rules of logic are very limited (natural deduction of Gentzen Von Plato (2008) employs ten primitive rules of proof) while the number of concepts used in a research domain reaches tens of thousands (e.g., MeSH ontology currently contains around 30,000 entry terms) and the number of factual statements that can be produced about complex systems (e.g. the human organism) is virtually unlimited. Since the application of logical Luo et al. (2023) and factual reasoning Mohri & Hashimoto (2024) includes the production of propositions about concepts, an adequate grasp of relevant concepts is a precondition for valid reasoning.

We observed a high correlation between conceptual integrity in biology, chemistry and medicine and the MMLU benchmark Hendrycks et al. (2021a). MMLU covers 57 subjects across STEM, the humanities, the social sciences and tests both world knowledge and problem solving ability on elementary and professional levels. It is important to note that the conceptual integrity and MMLU benchmarks evaluate complementary aspects of language understanding. Whereas MMLU addresses factual knowledge and problem-solving, the current benchmark focuses narrowly on the relations between conceptual constituents (labels, definitions and referents).

Complementary lines of work have explored conceptual understanding through the lens of representation geometry, including theoretical frameworks for concept bottleneck models Luyten & van der Schaar (2024), geometric analysis of categorical and hierarchical concepts in LLMs Park et al. (2025), and structural investigations of concepts via sparse autoencoder features Li et al. (2024). An orthogonal approach to conceptual understanding comes from cognitive science studies of semantic feature norms, where human participants generate attributes for everyday concepts McRae et al. (2005). Recent work has begun bridging this tradition with LLM capabilities, both by using LMs to generate semantic features Hansen & Hebart (2022) and by creating large-scale AI-enhanced feature norm datasets Suresh et al. (2025). These approaches are largely complementary: while feature norms capture graded semantic attributes from everyday concepts, the semantic integrity framework targets technical domains where expert-curated ontologies aspire to consistent definitions and verifiable ground truth.

Anomalies in the semantic field size (SFS) task performance highlighted fundamental differences in how ontologies structure concepts across scientific domains. Chemical ontologies such as ChEBI enumerate specific molecular structures as referents. For example, D-ribose lists thousands of distinct phosphorylated and modified variants, each representing a chemically distinguishable entity. In contrast, biological ontologies such as Gene Ontology and MeSH predominantly organize concepts into taxonomic hierarchies where "children" represent subclasses rather than individual instances.

Medical ontologies exhibit mixed patterns depending on the concept type (e.g., disease ontologies are class hierarchies while drugs refer to formulations of specific compounds). The SFS task instructions attempted to accommodate both interpretations by stating: "If the concept is a class consisting of subclasses, report the number of subclasses as the size of the semantic field. For concepts with physically countable distinct (non-identical) referents, report the corresponding estimate." However, this distinction proved ambiguous for biological concepts where both interpretations are semantically valid. A lymphocyte is simultaneously a class with 3 major subclasses and a physically countable entity with trillions of instances in vivo.

This finding has important implications for benchmark design across scientific domains. Tasks that perform well within a single domain may encounter systematic measurement challenges when applied cross-domain if the underlying ontological structures differ in fundamental ways. Specifically, the nature of the referent of a concept might depend on whether we are dealing with conceptual hierarchies that ultimately refer to physical entities (e.g., cell types and cells) or not (e.g., disease classifications). This does not invalidate the semantic-field-size task within the conceptual integrity framework but highlights that operationalizing this measurement may require domain-specific instructions and an understanding of the level of conceptual implication that is most relevant to a domain expert.

On a higher level, an apparent connection exists between conceptual integrity and compositional generalization which is the ability to understand novel combinations of known components systematically combined according to learned rules Lake & Baroni (2018). Similarly to the systematicity argument of Fodor and Pylyshyn Fodor & Pylyshyn (1988), our framework treats conceptual integrity as explicitly compositional: definitions are sets of decidable selection criteria, and adding or removing criteria creates more or less specific concepts in a hierarchy. This extends naturally to noun phrase compositionality where noun modifiers recruit additional selection criteria above and beyond those of the core noun. Crucially, this compositional structure must ultimately ground in elementary concepts to avoid infinite regress. We propose that these elementary concepts what some philosophers call qualia Tye (2021) have selection criteria rooted directly in perception (e.g., the concept 'blue' is defined simply as 'it is blue'). This grounding in perception provides the base case for recursively defined concepts and aligns with everyday usage where vague concepts are defined as 'I know it when I see it' Wikipedia contributors (2024). Our framework thus positions conceptual integrity as a foundational capability underlying both compositional generalization and systematic reasoning: accurate understanding of constituent concepts (including perceptually grounded elementary ones) is prerequisite to valid reasoning about their combinations.

## 5   LIMITATIONS

1. The proposed conceptual semantics framework is currently limited to conceptual categories/entities as the logical analysis and adoption of gold standards for predicates in open-ended domains such as biology is far from trivial Kilicoglu et al. (2011).

2. The subset of concepts included in the evaluation is small and by no means exhaustive. The main utility of the present work lies in providing a logical framework, proof-of-concept study and a starting point for more comprehensive benchmarks. However, the statistical power of our analyses was very high, as evidenced by extremely small FDR-corrected q-values (e.g., $q < 10^{-14}$ for identifying low-performing models, $q < 0.002$ for identifying high-performing models) and the significance of 26 out of 29 category-level effects after FDR correction, suggesting that increasing the concept selection in the chosen domains would not substantially alter our statistical conclusions.

3. The domains of interest were chosen based on the authors' perception of relevance and do not correspond to a general recommendation or a limitation of the framework.

4. Baseline data for referents is expected to be incomplete (ontologies are typically not exhaustive) and potentially inconsistent. It is likely that the benchmark is estimating the lower bound of performance on concepts with large semantic fields.

5. Provided tasks do not assess conceptual integrity in its purest form. The task formulations included in the benchmark require the following of instructions and production of syntactically correct JSON, both of which are not related to conceptual integrity but contribute to the performance significantly.

## ACKNOWLEDGMENTS

The authors would like to thank anonymous reviewers for comments and suggestions on earlier versions of the manuscript.

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

## A    SUPPLEMENTARY MATERIAL

An anonymized supplementary archive (`supplementary_material.zip`) contains concept annotations, prompt templates, LM responses to tests, scores from the LM judge, summary reports and plots, as well as the code used to generate them.

### A.1    FORMAL FRAMEWORK OF CONCEPTUAL SEMANTICS

#### A.1.1    CONCEPT

Concept $C$ is a set of selection criteria $C = \{c_1, c_2, ..., c_n\}$ for its referents $R_C$.

Let selection criterion $c_i$ be a proposition corresponding to a linguistic statement[1] about some entity $x$. Given an entity $r$ and selection criteria $C$, a decision function $d(r, C) \rightarrow \{true, false\}$ will output $true$ only if all selection criteria $c_i \in C$ apply to $r$. Thus, the application of $d$ to its arguments corresponds to a logical conjunction of the selection criteria $C$ when evaluated in the context of $r$:

$$d(r, C) \equiv \left( r \vdash \bigwedge_{c_i \in C} \right). \tag{1}$$

If $r$ satisfies the selection criteria $C$ then $r$ is considered a referent of $C$.

#### A.1.2    SEMANTIC FIELD

The set of referents $R_C$ of concept $C$ is a collection of imagined entities $R_C = \{r_1, r_2, ..., r_n\}$ that satisfy selection criteria C.

$$C \implies R_C : \forall r_i \in R_C.(d(r_i, C) = \text{true}) \tag{2}$$

In subsequent text we will refer to $R_C$ as the **semantic field** of concept $C$.

#### A.1.3    IDENTITY OF SELECTION CRITERIA

Selection criteria $c_i$ and $c_j$ are identical if and only if their symbolic representations (as linguistic terms) are identical.

$$(c_i = c_j) \Leftrightarrow (E(\{c_i\}) = E(\{c_j\})), \tag{3}$$

where function $E$ maps a set of selection criteria to a corresponding linguistic form in a suitable language (e.g. English).

#### A.1.4    EQUIVALENCE OF SELECTION CRITERIA

Selection criteria $c_i$ and $c_j$ are equivalent if and only if they imply the same set of referents

$$(c_i \equiv c_j) \Leftrightarrow (c_i \implies R_{c_i}) \wedge (c_j \implies R_{c_j}) \wedge (R_{c_i} = R_{c_j}). \tag{4}$$

One might also want to distinguish between the logical and practical aspects of the equivalence of selection criteria. Thus, two sets of selection criteria are logically equivalent if they imply each other. For example, given the formulations $C_1$ and $C_2$ of the concept *human protein coding gene* as:

$C_1 = \{c_1 : \text{"it is a human gene"}, c_2 : \text{"it encodes protein"}\}$

$C_2 = \{c_3 : \text{"it is a gene"}, c_4 : \text{"it is found in human"}, c_5 : \text{"it encodes protein"}\}.$

---

[1]Linguistic statements (e.g., "it will rain tomorrow") are the subset of linguistic expressions that can be true or false. Linguistic expressions (e.g., "white cat") are words and grammatically valid word combinations that are a subset of finite combinations of symbols from an alphabet.

One can safely state that $C_1 \Leftrightarrow C_2$ because $c_2 = c_5$ and $c_1 \Leftrightarrow c_3 \wedge c_4$.

It follows that two logically equivalent formulations even though not identical refer to the same set of referents in all possible contexts. On the other hand, it is also possible that two definitions that are not logically equivalent refer to the same referents in some context $R' \subset R$ but not necessarily in all possible subsets of $R$.

### A.1.5  LINGUISTIC INTERPRETATION

Linguistic interpretation is a chain of reference where linguistic term $t$ refers to concept $C$ that implies referents $R_C$.

$$t \implies C \implies R_C. \tag{5}$$

Thus, expression (5) establishes a sequence where a symbol refers to a concept that refers to its semantic field. The implication from symbol to concept is based on arbitrary association as evidenced by translatability between natural languages while the implication between a concept and its referents is determined by the selection criteria.

### A.1.6  CONCEPTUAL IDENTITY

Concepts $C_i$ and $C_j$ are identical if and only if they can be decomposed into equivalent selection criteria so that there exists a bijection between the sets of selection criteria $C_i$ and $C_j$ where $\{c_k, c_l\} : c_k \in C_i, c_l \in C_j$ imply the same set of referents.

$$(C_i = C_j) \Leftrightarrow \begin{array}{l} f : C_i \to C_j, f(g(c_l)) = c_l \\ g : C_j \to C_i, g(f(c_k)) = c_k, \end{array} \tag{6}$$

where $c_k \in C_i, c_l \in C_j, \forall r \in R.(d(r, \{c_k\}) = d(r, \{c_l\}))$.

### A.1.7  CONCEPTUAL EQUIVALENCE

Concepts $C_1$ and $C_2$ are equivalent if their semantic fields are identical.

$$\begin{array}{l} (C_i \equiv C_j) \Leftrightarrow (R_{C_i} = R_{C_j}) \\ (R_{C_i} = R_{C_j}) \Leftrightarrow \forall r \in R.(d(r, C_i) = d(r, C_j)) (\text{by 2}) \end{array} \tag{7}$$

We can imagine a concept $C_1 = \{c_1\}$ with one selection criterion which yields a set of referents $R_{C_1}$ and an equivalent concept $C_2 = \{c_2, c_3\}$ that yields an identical set of referents $R_{C_2} = R_{C_1}$.

### A.2  PROPERTIES OF THE CONCEPTUAL SEMANTICS FRAMEWORK

Proofs of the properties outlined below are provided as Lean 4 code in the supplementary archive. The assistance of GitHub Copilot was used to suggest tactics for the proofs.

### A.2.1  COMPOSITIONALITY OF CONCEPTS AND NOUN PHRASES

Compositionality of concepts enables addition and removal of selection criteria to specify and relax the concept by restricting and expanding its semantic field, correspondingly.

$$C' = C \cup \{c\} \implies R_{C'} \subseteq R_C (\text{by 1 and 2}) \tag{8}$$
$$C'' = C \setminus \{c\} \implies R_C \subseteq R_{C''} (\text{by 1 and 2}) \tag{9}$$

This additive nature of selection criteria supports conceptual hierarchies. For example, assertions "all surgeons are doctors" and "all doctors are not surgeons" imply that the referents of *surgeon* satisfy the selection criteria in *doctor*, while the referents of *doctor* do not satisfy all selection criteria in

*surgeon*. This implies that the selection criteria in *doctor* are a subset of selection criteria in *surgeon* creating a nested hierarchy between the concepts *doctor* and *surgeon*:

$$C_1 \subset C_2 \implies R_{C_2} \subset R_{C_1},$$

where $C_1$ and $C_2$ are the selection criteria of the concepts *doctor* and *surgeon*, correspondingly. The referents of *doctor* are a superset of the referents of *surgeon*.

We can use function $E$ to map from the semantic form of concept to a corresponding linguistic form e.g., in English:

$$E(C_1) = \text{"doctor"}$$
$$E(C_2) = \text{"surgeon"}$$

Productivity of language (the ability to combine linguistic terms into distinctly meaningful expressions) is another manifestation of the compositionality of concepts. Thus, the semantic field of the expression "a white rabbit with a black hat and a pocket watch" is an intersection of the semantic fields of the terms "white", "rabbit", "with a black hat", "with a pocket watch" when interpreted according to English grammar in the given context.

$$E(\{c_1, c_2, c_3\}) = \text{"a white rabbit with a black hat and a pocket watch"}$$

where

$$E(\{c_1\}) = \text{"it is a white rabbit"}$$
$$E(\{c_2\}) = \text{"it has a black hat"}$$
$$E(\{c_3\}) = \text{"it has a pocket watch"}$$

### A.2.2 COMPOSITIONALITY OF MEANING

Contradictory selection criteria result in a concept with an empty semantic field. Such concepts can be regarded as nonsensical or meaningless (as opposed to meaningful), because there is no context where they can have referents.

The size of semantic field $R_C$ of concept $C$ is null when there exists in $C$ at least one pair of selection criteria $c_i, c_j$ such that there are no referents $r$ that satisfy $c_i$ and $c_j$ simultaneously:

$$\exists c_i \exists c_j \in C : \forall r \in R, \neg(d(r, \{c_i\}) \land d(r, \{c_j\}))) \implies (R_C = \emptyset)(\text{by 1 and 2}) \tag{10}$$

### A.2.3 COMPOSITIONALITY OF LOGICAL OPERATIONS IN NATURAL LANGUAGE

It is plausible that semantic fields of size null as exemplified above have also practical implications. Concepts can be combined with logical operators in natural language to expand or restrict the semantic field of a phrase and it seems that the interpretation of the result is contingent on the anticipated semantic field size.

For example, combination of adjectives with "or" in a noun phrase implies the union of their semantic fields:

$$C_i \lor C_j \implies R_{C_i} \cup R_{C_j} \tag{11}$$

Example 1: the set of referents of "sweet or sour sauce" coincides with the union of the referents of "sweet sauce" and "sour sauce".

Combination of non-contradictory adjectives with "and" implies an intersection of their semantic fields:

$$C_i \land C_j \implies R_{C_i} \cap R_{C_j} \tag{12}$$

Example 2: the set of referents of "sweet and sour sauce" is the intersection of the sets of referents of "sweet sauce" and "sour sauce".

However, a conjunction of contradictory adjectives in a noun phrase is typically interpreted as a union operation.

Example 3: the set of referents of "hot and cold beverages" is the union of the sets of referents of "hot beverages" and "cold beverages".

Likewise, the semantic field of a conjunction of nouns (e.g. "diamonds and pearls", "Alice and Bob") is to be interpreted as the union of the arguments since the intersection (e.g. the set of referents satisfying both the selection criteria of *diamonds* and *pearls*) is anticipated to be null.

# B  SUPPLEMENTARY METHODS

## B.1  DATA SOURCING AND CURATION

Various web resources (Supplementary Table 9) and databases were used to retrieve definitions and referents for selected concepts from the domains of biology, chemistry and medicine. Selection criteria were composed by a human expert based on the canonical definition. All annotations were reviewed for consistency by a human expert but, typically, no attempt was made to supplement incomplete lists of referents from human-curated ontologies.

Example data record of concept with a flat set of referents:

```
{
    "concept": "neutral amino acid at physiological pH",
    "definition": "Amino acids with uncharged R groups or " \
      "side chains.",
    "selection_criteria": [
        "it is an amino acid",
        "it has uncharged R groups or side chains"
    ],
    "ontology_id": "D021542",
    "ontology": "MeSH",
    "referents": [
        "Asparagine (Asn)",
        "Cysteine (Cys)",
        "Glutamine (Gln)",
        "Methionine (Met)",
        "Serine (Ser)",
        "Threonine (Thr)"
    ],
    "domain": "chemistry"
}
```

Example data record of concept with a nested set of referents:

```
{
    "domain": "biology",
    "concept": "epidermal cell",
    "definition": "Cell of the outermost, non-vascular layer" \
      " (epidermis) of the skin.",
    "selection_criteria": [
        "it is a cell",
        "it is located in the outermost, non-vascular layer" \
        " of the skin"
    ],
    "ontology": "MeSH",
    "ontology_id": "D000078404",
    "referents": {
        "Melanocytes": {
```

```
918                "Melanosomes": {}
919            },
920            "Keratinocytes": {
921                "HaCaT Cells": {}
922            },
923            "Merkel Cells": {}
924        }
925    }
```

## B.2 Instruction Templates for Testing

The following prompt templates were contextualized with concept-specific information to create tasks for the conceptual integrity benchmark. Placeholders in the prompt template designated by terms in curly brackets (e.g. '{definition}') were replaced by the corresponding values from context variables before sending the prompt to an LM. All other context variables were ignored (not rendered into the prompt).

### B.2.1 Instruction template for task "decide-concept"

> You are a biology, organic chemistry and medical research expert. Your task is to output the closest conceptual category that includes all entities that comply with the definition given below. Provide a concise response consisting of the name of the concept in double quotes and nothing else.
>
> <example>
> Definition: "a country located on the Scandinavian peninsula"
> Concept: "Scandinavian country"
> </example>
>
> <example>
> Definition: "Any of various nonplacental mammals bearing young that suckle and develop after birth in the mother's pouch"
> Concept: "marsupial"
> </example>
>
> Definition: "{definition}"
> Concept: ...

### B.2.2 Instruction template for task "decide-concept-from-selection-criteria"

> You are a biology, organic chemistry and medical research expert. Your task is to output the largest conceptual category that includes only entities that comply with the definition given below. The definition is provided as a set of necessary and sufficient criteria for the members of the category. Make sure to pick a category that includes all entities that satisfy the mentioned criteria and no entities that do not. Provide a concise response consisting of the name of the concept in double quotes and nothing else.
>
> <example>
> Definition: ["it is a country", "it is located on the Scandinavian peninsula"]
> Concept: "Scandinavian country"
> </example>
>
> <example>
> Definition: ["it is a nonplacental mammal", "it bears young that suckle"]
> Concept: "marsupial"

</example>

Definition: "{selection_criteria}"
Concept: ...

### B.2.3 INSTRUCTION TEMPLATE FOR TASK "LIMITED-LIST-REFERENTS"

You are a biology, organic chemistry and medical research expert. List up to
{n} referents (examples/instances) of the concept "{concept}". Do not repeat
any referent. For each referent provide a canonical name (with abbreviated
identifier in parentheses if available). Output the list in JSON format as a list
of strings. Do not include any comments or explanations beside the JSON
output.

Referents (JSON): ...

### B.2.4 INSTRUCTION TEMPLATE FOR TASK "LIMITED-LIST-REFERENTS-FROM-SELECTION-CRITERIA"

You are a biology, organic chemistry and medical research expert. List up
to {n} referents (examples/instances) of the concept defined below. The
definition is given as a list of necessary and sufficient criteria that each
referent must satisfy. Do not repeat any referent. For each referent provide
a canonical name (with abbreviated identifier in parentheses if available).
Output the list in JSON format as a list of strings. Do not include any
comments or explanations beside the JSON output.

Definition: {selection_criteria}
Referents (JSON): ...

### B.2.5 INSTRUCTION TEMPLATE FOR TASK "DECIDE-REFERENTS"

Your task is to decide which of the entities listed below are referents (ex-
amples or instances) of the concept "{concept}". Reply by listing from the
original list only such entities that are referents of {concept}. Output the list
as a valid JSON list of strings. Do not include any comments or explanations
besides the correctly formatted JSON output.
<example>
Entities: ["Poland", "Mongolia", "Spain", "Finland", "Uruguay", "Norway"]
Concept: "nordic country"
Referents (JSON): ["Finland", "Norway"]
</example>
Entities: entities
Concept: "concept"
Referents (JSON): ...

### B.2.6 INSTRUCTION TEMPLATE FOR TASK "SEMANTIC-FIELD-SIZE"

Semantic field is the set of referents (instances or examples or "children")
for a concept. Your task is to estimate the size of the semantic field for a
given concept. If the concept is a class consisting of subclasses, report the
number of subclasses as the size of the semantic field. For concepts with
unbounded number of abstract or imagined yet distinguishable (by some
relevant parameter) referents respond with "unlimited". For concepts with
physically countable distinct (non-identical) referents, report the correspond-
ing estimate. Do not consider indistinguishable instances (such as molecules

or viral particles of a kind) as separate referents. Respond by providing a correctly formatted JSON object containing lower and upper bounds and a point estimate (based on currently available information) of the semantic field size. Do not provide any comments or elaborations on the response. Respond with a valid JSON. Provide the lower and upper bounds of semantic field size R according to the following scale:

| Size category | Explanation |
|---|---|
| 0 < R <= 1e1 | "R ranges from 1 to 10" |
| 1e1 < R <= 1e2 | "R ranges from 11 to 100" |
| 1e2 < R <= 1e3 | "R ranges from 101 to 1000" |
| 1e3 < R <= 1e4 | "R ranges from 1001 to 10000" |
| 1e4 < R <= 1e5 | "R ranges from 10001 to 100000" |
| 1e5 < R <= 1e6 | "R ranges from 100001 to 1000000" |
| 1e6 < R <= 1e9 | "R ranges from 1000001 to 1000000000" |
| 1e9 < R <= 1e12 | "R ranges from 1000000001 to 1000000000000" |
| 1e12 < R | "R is finite but larger than 1000000000000" |
| R → "infinity" | "R is unlimited" |

<examples>
Concept: human
Response (JSON): {"lower bound": "1e9", "upper bound": "1e12", "point estimate": "8e9"}

Concept: country
Response (JSON): {"lower bound": "1e2", "upper bound": "1e3", "point estimate": "195"}

Concept: table salt
Response (JSON): {"lower bound": "1", "upper bound": "10", "point estimate": "1"}

Concept: circle
Response (JSON): {"lower bound": "unlimited", "upper bound": "unlimited", "point estimate": "unlimited"}
</examples>

Concept: {concept}
Response (JSON): ...

## B.3 SCORING

Scoring of responses was performed using the LLM-as-a-judge approach with Gpt-4o acting as the judge. Importantly, scoring instructions did not include contextual information (e.g. concept definition or label) that was provided in the test prompt while including additional information about the ground truth that was not available at test time. This arrangement was used to make the scoring task as orthogonal as possible to the test. In effect, the scoring task was reduced to the evaluation of equivalence between conceptual entities which is a logically simpler task (a one to one mapping between concept labels) than inferring a concept from the definition (definition is a set of propositions) and enumeration of referents (mapping between concept and referents is typically one to $n$).

Instructions for scoring responses to the both 'decide-concept' tasks ask to identify whether concepts A and B are strictly equivalent in the sense that A implies B and vice versa without exception. The instruction template is contextualized using the concept from the test response and the ground truth.

Instructions for scoring the 'limited-list-referents' and 'decide-referents' tasks ask to compare a lists submitted by a student to a reference list and return overlapping and non-overlapping entities from the student's list. A match between entities does not have to be exact but an entity provided by the student must be strictly equivalent to at least one reference example in the sense that it implies the corresponding reference example and vice versa without exception. All overlapping entities are

scored as true positives and the rest as false positives. For baseline lists of referents exceeding 72 items, the authoritative list in the prompt was constructed by retrieving $k = 3$ most similar referents to each student's response based on shortest embedding distance. Text embeddings for all referents were obtained using the "gte-large-en-v1.5" model Alibaba NLP (2025).

Estimations of semantic field size were scored using a simple algorithm (implemented in Python) that rewarded for each correctly guessed bound for the number of referents and penalized for every order of magnitude that the point estimate deviated from the ground truth.

In tasks with a binary outcome (two variations of 'decide-concept' task), average accuracy represents the ratio of correct answers to all answers across the studied concepts. In open-ended tasks that require the listing of referents (two variations of 'limited-list-referents' task), accuracy corresponds to the ratio of response entities overlapping with the ground truth list i.e. $TP/(TP + FP)$ where false positives (FP) are response entities not found in the ground truth. In the 'decide-referents' task, accuracy corresponds to $TP/(TP + FP + FN)$ where false negatives (FN) are entries from the ground truth list that were not found in the response. In the 'semantic-field' size test, response accuracy was calculated as $\frac{w*BN+w-PD}{2w}$ where

$BN = B/2$ and B is the number of correctly guessed bounds (e.g. BN=1 if both upper and lower bound were guessed correctly) and

$PD$ is the deviation of the point estimate from the ground truth in orders of magnitude (PD was capped at $w = 3$).

### B.3.1 INSTRUCTION TEMPLATES FOR SCORING

Instruction template for scoring matches between two lists of referents (**"score-referents"**):

> You are a biology, medicine and organic chemistry teacher. You will compare a reference list of entities to a list of entities produced by a student. Please output a JSON object where under the key "matches" you provide as a list of strings all entities from the student's list that match with the reference examples and under "mismatches" you list all entities that are not contained in the reference list. A match does not have to be exact but the entity must be strictly equivalent to at least one reference example in the sense that the entity given in the student's response implies the corresponding reference example and vice versa without exception. List only unique matches (i.e. make sure not to repeat any matches). Do not provide any explanations or text other than the properly formatted JSON object containing two lists of strings under keys "matches" and "mismatches".
>
> <example>
> Reference list: ["Sweden", "Norway", "Denmark"]
> Response submitted by the student: ["Finland", "Norway", "Spain", "Denmark"]
> Judgement (JSON): {"matches": ["Norway", "Denmark"], "mismatches": ["Finland", "Spain"]}
> </example>
>
> <example>
> Reference list: ["Estonia", "Latvia", "Lithuania"]
> Response submitted by the student: [{"country": "Finland", "short name": "Fin"}, {"country": "Latvia", "short name": "Lat"}, {"country": "Lithuania", "short name": "Lit"}]
> Reponse (JSON): {"matches": ["Latvia", "Lithuania"], "mismatches": ["Finland"]}
> </example>

```
    Authoritative information:
    {baseline}

    List submitted by the student:
    {student_response}

    Response (JSON): ...
```

Instruction template for scoring a match between two concepts **"concept-equivalence"**:

```
    You are a biology, organic chemistry and medical research expert. Your task
    is to decide whether the concepts A and B given below are strictly equivalent
    in the sense that A implies B and vice versa without exception. Respond in
    JSON format based on the examples below.

    <example>
    A: "rain"
    B: "rainfall"
    Response (JSON): "equivalent": true, "reason": "rain and rainfall are
    synonyms"
    </example>

    <example>
    A: "vehicle"
    B: "car"
    Response (JSON): "equivalent": false, "reason": "car is a vehicle but not all
    vehicles are cars"
    </example>

    A: "{conceptA}"
    B: "{conceptB}"
    Response (JSON): ...
```

## B.4 EXTERNAL RANKINGS

External rankings were obtained from OpenAI (2025) and Abdin et al. (2024). Correlations between
internal and external benchmarks were calculated on a subset of models that had been evaluated on
all benchmarks (10 models in total: claude-3-5-sonnet, claude-3-opus, gpt-4, gpt-4o, gpt-4o-mini,
llama3-70b-instruct, llama3-8b-instruct, o1-mini, phi-v4, qwen-v2.5-14b-instruct).

## B.5 STATISTICAL ASSESSMENT OF DEVIATIONS FROM EXPECTED PERFORMANCE

Hypergeometric distribution models the sampling of objects (without replacement) from a binary
pool (e.g., correct/incorrect responses) and estimates the probability of getting the observed number
of type I objects (e.g., correct responses) under the assumption of independence. The appeal of the
hypergeometric distribution lies in its intuitive modeling of the sampling procedure and minimal
number of associated assumptions. In our setting, we pooled accuracy estimates from all tasks except
the 'semantic-field-size' and all models to yield the total response pool of 14,490 responses including
9,063 correct ones. Based on the scoring procedures outlined above, each response could yield a
maximum score of 1 and a minimum of 0 making the summation of scores across tests an unbiased
operation. Given that all models were subjected to an equal number of trials, the task was to estimate
the probability of obtaining the observed number of correct responses or a more extreme result for
the given model in 966 attempts. We estimated both the upper and lower tail probability to highlight
models that deviated from the group performance at both ends. P-values were adjusted for multiple
testing using the FDR under dependence method by Benjamini & Yekutieli (2001).

## B.6 QUANTITATIVE ANALYSIS OF CONCEPT CATEGORIES

To address whether specific concept characteristics systematically influence model performance, we analyzed response accuracy across five categorization dimensions of concepts.

**Abstraction level** distinguished between concepts with physical/concrete referents (e.g., molecular structures and drugs), abstract entities (e.g., disease categories and cell types), and mixed referents (implication hierarchies containing both abstract and physical entities e.g., chemical compound ontologies). **Semantic field size** was categorized into small ($|R_C| < 50$), medium ($50 \leq |R_C| \leq 500$), and large ($|R_C| > 500$) tiers based on gold-standard referent counts. **Domain** classified concepts according to their scientific field (biology, chemistry, medicine). **Ontological structure** differentiated between concepts with a flat set of referents (direct children only, depth $\leq 1$) and hierarchically organized referents (nested subcategories, depth $\geq 2$). **Count of selection criteria** distinguished between concepts with a few (1-2) selection criteria or more (3-6).

Abstraction level classification was decided based on domain-specific rules applied to definitions and referent structures. In biology, referents of cell types were classified as abstract (referring to categories rather than countable individuals) while referents of genes and proteins were concrete (specific isolatable molecules). Chemistry concepts with hierarchical referent ontologies (depth $\geq$ 2) were classified as mixed, since they contain both compound class names and specific molecular structures. Some medical concepts referred to concrete entities (e.g., drug classes pointing to specific medications) while others did not (e.g., disease classifications). In general, explicit categorical language and hierarchical referent structure was associated with abstract concepts while flat lists of physical entities were indicative of concrete concepts.

For each task, we computed mean response accuracy for each instance (level) of a category and performed statistical comparisons using Mann-Whitney U tests (for binary categories) or Kruskal-Wallis H test (categories with more than two levels). We applied false discovery rate correction (Benjamini-Yekutieli procedure) and calculated effect size ratios as the ratio of best-performing to worst-performing category instance means. Since only 3 out of 29 category effects on task-specific performance were insignificant after FDR correction, we considered effects practically significant when the effect size ratio exceeded 1.5 (indicating $\geq$50% relative improvement in performance between category levels).

### B.6.1 ABSTRACTION LEVEL

Abstraction level showed the strongest and most consistent effects, with all six tasks exhibiting practically significant differences (effect size ratios 1.55–1.78$\times$, all $q < 10^{-35}$). However, the direction of effect varied systematically by task type. Abstract and mixed concepts led to better concept identification from definition or selection criteria than concrete concepts (decide-concept: 1.61$\times$, decide-concept-from-selection-criteria: 1.62$\times$). This likely reflects that abstract concepts have more linguistically explicit definitions, while concrete molecular structures may require chemical knowledge not captured in text-based definitions. On the other hand, concepts with concrete referents triggered more accurate referent enumeration than those with abstract or mixed referents (limited-list-referents: 1.73$\times$ and limited-list-referents-from-selection-criteria: 1.78$\times$).

### B.6.2 SEMANTIC FIELD SIZE

Semantic field size exhibited the largest effect size overall with the enumeration of referents for concepts with small semantic fields outperforming large ones. This pattern likely reflects both computational difficulty (retrieving 10 referents from a set of 50 versus 50,000) and knowledge coverage (smaller concepts tend to be more specialized and better documented). For concept identification tasks (decide-concept and decide-concept-from-selection-criteria) the opposite pattern was true indicating that it was easier for the models to identify the label for concepts with large semantic fields based on a definition or selection criteria.

### B.6.3 DOMAIN

Domain effects were moderate but consistent. Chemistry concepts showed lowest performance on four of six tasks, with particularly strong deficits in referent-related tasks (decide-referents: $1.54\times$ worse than biology; limited-list-referents-from-selection-criteria: $1.51\times$ worse than biology; semantic-field-size: $1.58\times$ worse than medicine). This pattern suggests that chemical nomenclature, structural formulas, and systematic naming conventions present unique challenges not fully captured in text-based training data. It also aligns with anecdotal observations from subject matter experts regarding the knowledge gaps of language models in chemistry relative to biology, for example. Biology concepts generally performed best, likely reflecting both the prevalence of biological discussions in web text and the more linguistically explicit nature of biological definitions compared to chemical nomenclature.

### B.6.4 ONTOLOGICAL STRUCTURE

Ontological structure showed opposing effects depending on task requirements. Concepts with hierarchical referent structures were associated with better concept identification from definitions (decide-concept: $1.60\times$, decide-concept-from-selection-criteria: $1.46\times$) while flat structures performed better in referent enumeration tasks (limited-list-referents: $1.43\times$, limited-list-referents-from-selection-criteria: $1.49\times$). Hierarchical referent structure might be related to richer semantic context around the concepts in the training data but it is difficult to assess.

### B.6.5 SELECTION CRITERIA

The number of selection criteria (defining features) showed negligible effects across all tasks (effect size ratios $1.01$–$1.14\times$, only 2 of 6 tests significant after FDR correction). It suggests that models handle definitions decomposable into 1-6 selection criteria with similar effectiveness implying adequate compositional understanding at the level of conjunctive feature combinations.

### B.7 SYSTEMATIC INVESTIGATION OF SEMANTIC FIELD SIZE TASK ANOMALIES

We calculated the normalized discrepancy between model estimates and gold standard semantic field sizes to investigate the anomalies.

$$\text{Normalized Discrepancy} = \frac{\text{abs}(|\hat{R_C}| - |R_C|)}{|R_C|} \tag{13}$$

where $|\hat{R_C}|$ is a point estimate of semantic field size for concept $C$ from a language model and $|R_C|$ is the gold standard.

First, we tested the hypothesis that SFS task performance inconsistency might be due to the incompleteness of the expert-curated gold standard. As exhaustive enumeration of referents is much harder for concepts with thousands rather than dozens of referents, one would expect to see higher normalized discrepancy for concepts with larger semantic fields. To that end, we partitioned concepts by semantic field size into small ($|R_C| < 50$), medium ($50 <= |R_C| <= 500$) and large semantic field groups ($|R_C| > 500$). Mann-Whitney U test of whether the discrepancy was higher in large vs small semantic fields was highly insignificant (p = 1.000). Spearman correlation between semantic field size and normalized discrepancy was signicantly negative ($\rho$ = -0.178, p = 6.105e-21), indicating that point estimates for concepts with larger semantic fields exhibited marginally better agreement with gold standard. These results contradict the incompleteness hypothesis, suggesting that gold standard quality is unlikely to be the primary issue behind inconsistencies.

When we ordered concepts with a small semantic field by descending mean discrepancy, 10 of the top 12 were cell types. Some SFS estimates differed by as much as 27 orders of magnitude from the gold standard suggesting that there was a problem with task comprehension rather than model performance or gold standard. Specifically, the task instruction "estimate the size of the semantic field" proved fundamentally ambiguous for biological cell type concepts where models reasonably interpreted the question as requesting counts of physical instances (e.g., billions of lymphocytes in the human body) while gold standards from GO/MeSH ontologies enumerate functional subclasses

(e.g., 3 major lymphocyte types: T cells, B cells, NK cells). For more information around this failure mode see section 3.5.2.

Table 5: **Effect size ratios comparing best- to worst-performing instances of concept categories across tasks.** Values $\geq 1.5$ (bolded) indicate practically significant effects where the best category performs at least 50% better than the worst. All comparisons shown were statistically significant after FDR correction ($q < 0.05$).

| Task | Category | Effect Size Ratio | Best | Worst | $q$-value |
|------|----------|-------------------|------|-------|-----------|
| *Overall (pooled across all tasks)* | | | | | |
| | Semantic field | 1.35 | small | large | $< 10^{-135}$ |
| | Abstraction level | 1.33 | abstract | mixed | $< 10^{-131}$ |
| | Domain | 1.23 | biology | chemistry | $< 10^{-80}$ |
| | Ontology structure | 1.14 | flat | hierarchical | $< 10^{-27}$ |
| | Selection criteria | 1.10 | 3–6 | 1–2 | $< 10^{-15}$ |
| *decide-concept* | | | | | |
| | Abstraction level | **1.61** | mixed | concrete | $< 10^{-34}$ |
| | Ontology structure | **1.60** | hierarchical | flat | $< 10^{-38}$ |
| | Semantic field | 1.47 | large | small | $< 10^{-22}$ |
| | Domain | 1.30 | chemistry | biology | $< 10^{-13}$ |
| *decide-concept-from-selection-criteria* | | | | | |
| | Abstraction level | **1.62** | abstract | concrete | $< 10^{-35}$ |
| | Ontology structure | 1.46 | hierarchical | flat | $< 10^{-27}$ |
| | Semantic field | 1.38 | large | small | $< 10^{-17}$ |
| *decide-referents* | | | | | |
| | Abstraction level | **1.55** | abstract | mixed | $< 10^{-77}$ |
| | Domain | **1.54** | biology | chemistry | $< 10^{-70}$ |
| | Semantic field | 1.49 | medium | large | $< 10^{-73}$ |
| *limited-list-referents* | | | | | |
| | Semantic field | **2.05** | small | large | $< 10^{-100}$ |
| | Abstraction level | **1.73** | concrete | mixed | $< 10^{-133}$ |
| | Domain | **1.50** | biology | chemistry | $< 10^{-75}$ |
| | Ontology structure | 1.43 | flat | hierarchical | $< 10^{-77}$ |
| *limited-list-referents-from-selection-criteria* | | | | | |
| | Semantic field | **2.18** | small | large | $< 10^{-100}$ |
| | Abstraction level | **1.78** | concrete | mixed | $< 10^{-127}$ |
| | Domain | **1.51** | biology | chemistry | $< 10^{-69}$ |
| | Ontology structure | 1.49 | flat | hierarchical | $< 10^{-83}$ |
| *semantic-field-size* | | | | | |
| | Abstraction level | **1.62** | concrete | mixed | $< 10^{-55}$ |
| | Domain | **1.58** | medicine | chemistry | $< 10^{-47}$ |
| | Ontology structure | 1.47 | flat | hierarchical | $< 10^{-48}$ |
| | Semantic field | 1.26 | medium | large | $< 10^{-11}$ |

Table 6: **Distribution of concepts across categorization dimensions in the benchmark dataset.**

| Dimension | Category | Count |
|---|---|---|
| Abstraction level | Concrete | 56 |
| | Abstract | 55 |
| | Mixed | 76 |
| Semantic field size | Small ($<$50) | 65 |
| | Medium (50–500) | 74 |
| | Large ($>$500) | 48 |
| Domain | Biology | 55 |
| | Chemistry | 89 |
| | Medicine | 43 |
| Ontology structure | Flat | 56 |
| | Hierarchical | 131 |
| Selection criteria | Few (1–2) | 145 |
| | Several (3–6) | 42 |

## C  EXAMPLES OF FAILURE MODES

### C.1  CONCEPT ANNOTATION FAILURE

```
{
    "domain": "biology",
    "concept": "antioxidant system protein",
    "ontology": "GO",
    "ontology_id": "GO:0016209",
    "definition": "A protein with antioxidant activity.",
    "selection_criteria": [
        "it is a protein",
        "it has antioxidant activity"
    ],
    "referents": [...]
}
```

### C.2  LANGUAGE MODEL FAILURE

Excerpts of responses to an identical query from Gpt-4o and Gpt-4o-mini:

```
"model": "gpt-4o",
"response": [
    "Adenosine 3',5'-cyclic monophosphate (cAMP)",
    "Guanosine 3',5'-cyclic monophosphate (cGMP)",
    "Cytidine 3',5'-cyclic monophosphate (cCMP)",
    "Uridine 3',5'-cyclic monophosphate (cUMP)",
    "Inosine 3',5'-cyclic monophosphate (cIMP)",
    "Thymidine 3',5'-cyclic monophosphate (cTMP)",
    ...
]

# --------- VS ---------

"model": "gpt-4o-mini",
"response": [
    "Adenosine monophosphate (AMP)",
    "Cytidine monophosphate (CMP)",
```

```
    "Guanosine monophosphate (GMP)",
    "Uridine monophosphate (UMP)",
    "2'-Deoxyadenosine monophosphate (dAMP)",
    "2'-Deoxycytidine monophosphate (dCMP)",
  ...
  ]
```

Excerpts of responses to an identical query from Claude 3 Sonnet and Haiku:

```
  "model": "claude-3-sonnet",
  "response": [
      "Cyclic adenosine monophosphate (cAMP)",
      "Cyclic guanosine monophosphate (cGMP)",
      "Cyclic cytidine monophosphate (cCMP)",
      "Cyclic uridine monophosphate (cUMP)",
      "Cyclic inosine monophosphate (cIMP)",
      "Cyclic xanthosine monophosphate (cXMP)",
      ...
  ]

  # --------- VS ---------

  "model": "claude-3-haiku"
  "response": [
      "Adenosine Monophosphate (AMP)",
      "Guanosine Monophosphate (GMP)",
      "Cytidine Monophosphate (CMP)",
      "Thymidine Monophosphate (TMP)",
      ...
  ]
```

Excerpts of responses to an identical query from Llama 3 70B and 8B:

```
  "model": "llama3-70b-instruct",
  "response": [
    "cyclic AMP (cAMP)",
    "cyclic GMP (cGMP)",
    "cyclic IMP (cIMP)",
    "cyclic CMP (cCMP)",
    "cyclic UMP (cUMP)",
    "cyclic AMP-PS (cAMP-PS)",
  ...
  ]

  # --------- VS ---------

  "model": "llama3-8b-instruct",
  "response": [
      "ATP (Adenosine Triphosphate)",
      "GTP (Guanosine Triphosphate)",
      "CTP (Cytidine Triphosphate)",
      "UTP (Uridine Triphosphate)",
  ...
  ]
```

## C.3 TASK INSTRUCTION AMBIGUITY: LYMPHOCYTE EXAMPLE

The following example demonstrates the systematic ambiguity in the semantic field size task when applied to biological cell type concepts:

```
{
    "domain": "biology",
    "concept": "lymphocyte",
    "gold_standard_referents": 3,
    "referents": ["T cell", "B cell", "NK cell"]
}

Model responses (examples):
- GPT-4o: "point_estimate": "100000000000"
- Claude-3-Sonnet: "point_estimate": "2000000000000"
- Llama3-70B: "point_estimate": "5000000"
```

# D    SUPPLEMENTARY TABLES AND FIGURES

Table 7: Overview of concepts included in the benchmark dataset.

|  | Chemistry | Biology | Medicine | Overall |
|---|---|---|---|---|
| Concepts | 89 | 55 | 43 | 187 |
| Referents | 778,189 | 30,755 | 8,531 | 817,475 |
| Referents per concept (average) | 8,744 | 559 | 198 | 4,372 |
| Referents per concept (max) | 151,204 (carbonyl compound) | 20,788 (human protein coding gene) | 548 (cholinesterase inhibitor) | 151,204 |
| Referents per concept (min) | 3 (nitrogen oxide) | 3 (adipocyte) | 10 (selective serotonin uptake inhibitor) | 3 |

Table 8: Number of concepts per domain that were subjected to each test

| Test | Chemistry | Biology | Medicine |
|---|---|---|---|
| decide-concept | 89 | 55 | 43 |
| decide-concept-from-selection-criteria | 89 | 55 | 43 |
| limited-list-referents | 89 | 55 | 43 |
| limited-list-referents-from-selection-criteria | 89 | 55 | 43 |
| semantic-field-size | 89 | 55 | 43 |
| decide-referents | 46 | 8 | 10 |

Table 9: List of public web sources that were repeatedly used to gather information about concepts.

| Source | Count |
|---|---|
| https://www.ebi.ac.uk/chebi | 114 |
| https://www.ncbi.nlm.nih.gov/mesh | 62 |
| https://en.wikipedia.org/wiki | 13 |
| http://geneontology.org | 9 |
| https://www.cancer.gov/about-cancer/treatment/drugs | 7 |
| https://reactome.org | 5 |
| https://amigo.geneontology.org/amigo/search | 4 |
| https://www.ncbi.nlm.nih.gov/gene | 2 |
| https://pmc.ncbi.nlm.nih.gov/articles | 2 |

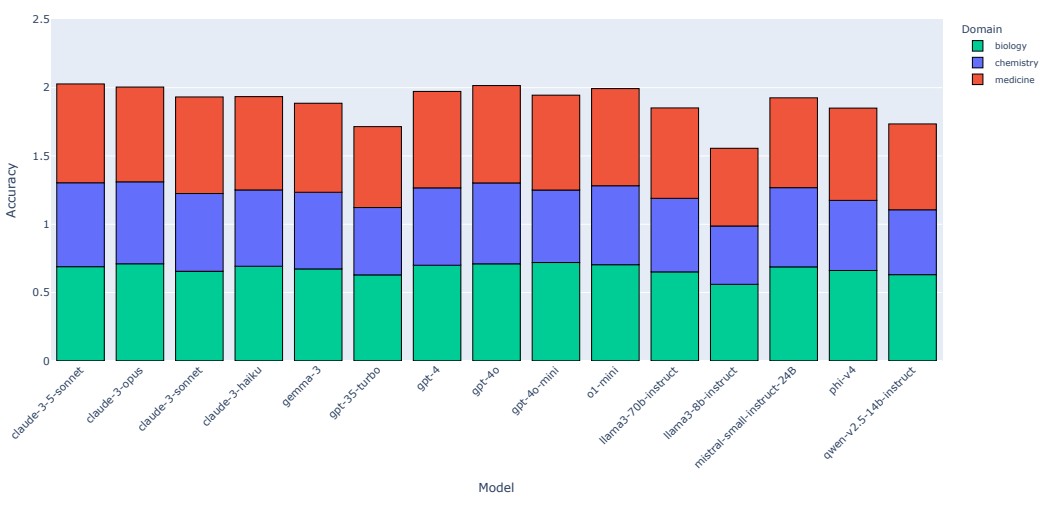

Figure 4: Performance of models across domains.

