# OpenReview forum: "Probing the Boundaries of Concepts in Language Models"
_ICLR.cc/2026/Conference — Submitted to ICLR 2026_

### Official Review · Reviewer_gJfc · 2025-10-20

**Soundness:** 4
**Presentation:** 3
**Contribution:** 3
**Rating:** 8
**Confidence:** 4

**Summary:**

The authors investigate concept understanding in some scientific domains. Information about a concept is scattered across documents, and the paper asks whether an LLM can synthesize across sources to form a complete model of a concept. They define a framework for what constitutes a concept, given by its label, definition/selection criteria, and referents. They also construct a benchmark on concept understanding by selecting several scientific concepts and defining a series of related tasks.

**Strengths:**

I think that the contribution of the benchmark and framework are valuable and timely. I especially see this work as useful toward the goal of evaluating the use of LLMs in producing novel scientific research.

The definition of concepts based on label/definition/referents is interesting and novel (or at least, I have not heard of this idea). The definitions are formalized with some basic equivalence and compositionality properties.

While the benchmark the authors have constructed is limited in scope, their framework and task definitions can be used to construct similar benchmarks for other domains. The tasks in the benchmark have a direct relationship to the theoretical framework, while also having obvious concrete applications to users of LLMs.

The scope of the tasks is precisely defined, so a cross-comparison of performance between tasks is a meaningful test of ‘conceptual integrity’, as the authors put it.

**Weaknesses:**

I somewhat disagree with the presentation of this as a general investigation of conceptual understanding. I see it as a benchmark of scientific concept understanding (or even more specifically understanding of concepts in drug discovery). In my opinion, the understanding of general, everyday concepts is a separate line of inquiry that is not addressed here.

I think conceptual understanding is a very broad and active line of work, with several threads that the authors do not mention. For example, I am thinking of work in modelling concepts via their representations/representation geometry, e.g.:

Luyten and van der Schaar. "A theoretical design of concept sets: improving the predictability of concept bottleneck models."  (NeurIPS 24)

Park et al., "The Geometry of Categorical and Hierarchical Concepts in Large Language Models" (ICLR 25)

Li et al.,  "The Geometry of Concepts: Sparse Autoencoder Feature Structure "

I’m not at all implying that the results should be more general – I would just suggest that the scope of the paper and its relation to prior work on concept understanding be defined more precisely.


While most of the paper was easy to follow, I had a few points of confusion. The last paragraph of Section 4 was difficult to understand - I am not sure if you are suggesting that there is some fundamental issue with categorizing biological concepts with hierarchies, or if there is just an error in the source data.

I also did not really understand why the semantic field size task was included - it seems to me from the discussion that the ground truth used to evaluate the responses was very inaccurate. In this case, what does the task tell us?

**Questions:**

How were the referents chosen? I assume based on the fact that some concepts have 100k+ referents that there is an automatic method.

What is the meaning of point 5 on page 2, semantic compositionality on proposition level? It is mentioned as an aspirational goal, and I’m curious what you see as the gap/future improvement to the framework.

Have you compared with performance on datasets restricted to biology/chemistry/medicine domains (e.g. in MMLU or GPQA the questions are labelled with the subject)? I would suspect that the correlation would be higher this way.

---

> ### Author Response · Authors · 2025-11-19
> **Response to Weakness 1**
>
> Weakness 1: "I somewhat disagree with the presentation of this as a general investigation of conceptual understanding. I see it as a benchmark of scientific concept understanding (or even more specifically understanding of concepts in drug discovery). In my opinion, the understanding of general, everyday concepts is a separate line of inquiry that is not addressed here.
>
> I think conceptual understanding is a very broad and active line of work, with several threads that the authors do not mention. For example, I am thinking of work in modelling concepts via their representations/representation geometry, e.g.:
>
> Luyten and van der Schaar. "A theoretical design of concept sets: improving the predictability of concept bottleneck models." (NeurIPS 24)
>
> Park et al., "The Geometry of Categorical and Hierarchical Concepts in Large Language Models" (ICLR 25)
>
> Li et al., "The Geometry of Concepts: Sparse Autoencoder Feature Structure "
>
> I’m not at all implying that the results should be more general – I would just suggest that the scope of the paper and its relation to prior work on concept understanding be defined more precisely."
>
> RESPONSE:
>
> We completely agree with this important distinction and thank the reviewer for the clear articulation. Our work indeed focuses on **technical scientific concepts** rather than general everyday concepts, and we will revise the presentation to make this scope explicit from the outset.
>
> The reviewer correctly identifies fundamental differences between technical/scientific concepts (discrete boundaries, expert consensus definitions, enumerable referents, formal specifications) and everyday concepts (graded membership, prototype effects, usage-based definitions, context-dependent meaning). This focus on technical concepts was a deliberate methodological choice driven by the availability of authoritative ground truth (expert-curated databases: ChEBI with 195K+ entities, MeSH with 30K+ terms, Gene Ontology) that enables reliable baseline evaluation unavailable for everyday concepts.
>
> While our **application** targets technical scientific domains, the **framework itself is domain-agnostic**—it applies to any concept with: (1) a linguistic label, (2) selection criteria (definition), and (3) enumerable referents. Extension to other technical domains with expert consensus should be straightforward. Extension to everyday concepts, on the other hand, is not straightforward due to the logical inconsistencies in the semantics of feature norms (they represent an assortment of semantically distinct statement types such as facts, category membership assertions or necessary criteria), prototype effects and graded category membership.
>
> As a whole, we do not see this as a significant shortcoming because knowledge domains where semantic precision is critical are expected to benefit most from the applications of the proposed framework (e.g., from principled semantic benchmarking of LMs and rigorous semantic analysis).
>
> We propose the following amendments to the manuscript:
>
> - **Location:** Abstract, line ~3 (page 1)
> - **Previous text:**
>
> >Systematic investigation of conceptual understanding in language models has not received much attention. This gap can be bridged by a formalized theory of conceptual semantics that maps naturally to instruction templates for natural language agents. We propose a simple framework expressible in first-order logic to address the semantic compositionality of concepts, noun phrases and conceptual hierarchies. The framework is used to derive a conceptual integrity benchmark with 6 tasks that are applied to a selection of 187 concepts from the domains of biology, chemistry and medicine.
>
> - **New text:** (lines 11-18 in the updated pdf)
>
> >Systematic investigation of the understanding of **scientific** concepts has not received much attention in language models. This gap can be bridged by a formalized theory of conceptual semantics that maps naturally to instruction templates for natural language agents. We propose a simple framework expressible in first-order logic to address the semantic compositionality of **scientific** concepts, noun phrases and conceptual hierarchies. The framework is used to derive a conceptual integrity benchmark with 6 tasks that are applied to a selection of 187 concepts from the domains of biology, chemistry and medicine.
>
>
> ---
>
> - **Location:** Section 1 (Introduction)
> - **Added text:** (lines 40-45 in the updated pdf)
>
> >We focus specifically on concepts from biology, chemistry, and medicine—concepts with well-defined boundaries, expert-curated definitions, and enumerable referents. This contrasts with everyday concepts which exhibit graded category membership, prototype effects, and lack authoritative consensus annotations. While our theoretical framework is domain-agnostic, the associated benchmark targets technical and scientific domains where ground truth can be reliably established.

---

> ### Author Response · Authors · 2025-11-19
> **Amendments continued**
>
> ---
>
> - **Location:** Section 4 (Discussion and Related Work)
> - **Previous text:**
> >We have proposed a logical framework of conceptual semantics that includes concept labels, definitions and referents. Based on this framework, a conceptual integrity benchmark was designed and applied to measure the degree of correspondence between LM responses and baseline information from canonical sources (e.g. databases, ontologies and research papers). The external validity of the benchmark is demonstrated by the consistent degradation of conceptual integrity measure in smaller instances of the same model.
>
> - **New text:** (lines 435-438 in the updated pdf)
> >We have proposed a logical framework of the **compositional semantics of scientific concepts** that includes concept labels, definitions and referents. Based on this framework, a conceptual integrity benchmark was designed and applied to measure the degree of correspondence between LM responses and baseline annotations from sources representing **expert-curated consensus** (e.g. databases, ontologies and research papers). The external validity of the benchmark is demonstrated by the consistent degradation of conceptual integrity measure in smaller instances of the same model.
>
>
> ---
>
> - **Location:** Section 4 (Discussion and Related Work)
>
> - **New text:** (lines 470-473 in the updated manuscript)
> >Complementary lines of work have explored conceptual understanding through the lens of representation geometry, including theoretical frameworks for concept bottleneck models (Luyten & van der Schaar, 2024), geometric analysis of categorical and hierarchical concepts in LLMs (Park et al., 2025), and structural investigations of concepts via sparse autoencoder features (Li et al., 2024).

---

> ### Author Response · Authors · 2025-11-20
> **Response to Weakness 2**
>
> Weakness 2: "While most of the paper was easy to follow, I had a few points of confusion. The last paragraph of Section 4 was difficult to understand - I am not sure if you are suggesting that there is some fundamental issue with categorizing biological concepts with hierarchies, or if there is just an error in the source data."
>
> RESPONSE:
>
> The expert-curated ontologies that we used to annotate concepts (e.g. ChEBI and MeSH) are highly regarded expert-curated knowledge bases for chemistry and biomedical research, respectively. Typically, researchers do not question the comprehensiveness and consistency of these data sources. Recently, in terms of knowledge coverage and perhaps even consistency, best performing LMs seem to be approaching the same scale (although we have not seen comprehensive benchmarking on this). We just wanted to note that, as is expected for any large scale annotation effort, occasional inconsistency or incompleteness is expected even in the expert-curated gold standard and hinted that LMs may provide a way to identify some of these issues.

---

> ### Author Response · Authors · 2025-11-20
> **Response to Weakness 3**
>
> Weakness 3: "I also did not really understand why the semantic field size task was included - it seems to me from the discussion that the ground truth used to evaluate the responses was very inaccurate. In this case, what does the task tell us?"
>
> RESPONSE:
>
> We appreciate this question. The SFS task was included because our framework identifies semantic field size as a core feature of conceptual integrity. However, you raise a valid concern about the gold standard accuracy.
>
> The weak correlation (0.03-0.26) between SFS and other tasks is indeed anomalous. This suggests either: (1) models behave inconsistently on SFS compared to other tasks, or (2) the gold standard has systematic limitations.
>
> We propose a testable hypothesis: if gold standard incompleteness is the primary issue, then the discrepancy between LM estimates and gold standard values should increase with semantic field size. That would imply a relatively lower correlation between SFS estimates and gold standard for concepts with a large semantic field as concepts with hundreds of referents (e.g., types of enzymes) are more prone to incomplete annotation in the gold standard than concepts with dozens of referents (e.g., types of blood cells).
>
> Please see response "Investigation of SFS task anomalies" below, for an update on this topic.

---

> > ### Author Response · Authors · 2025-11-25
> > **Investigation of SFS task anomalies**
> >
> > We have completed a comprehensive analysis investigating SFS task anomalies (new Appendix Section "Systematic Investigation of Semantic Field Size Task Anomalies").
> >
> > ### Added to manuscript Results section (lines 291-295):
> >
> > > "To investigate whether model failure, gold standard incompleteness or instruction ambiguity explains these anomalies, we conducted a detailed analysis comparing normalized discrepancies between point estimates and gold standard semantic field sizes depending on the magnitude of concept's semantic field (Supplementary Section~\ref{sec:sfs_analysis}). This analysis revealed that a major issue was task instruction ambiguity arising from fundamental differences in ontological structure across domains, particularly for biological cell type concepts where models confused class enumeration with physical instance counting. For example, when asked about "lymphocyte" semantic field size, models often responded with estimates in the billions or trillions (reflecting the total number of lymphocytes in the human body) rather than 3-5 (the number of major lymphocyte subtypes listed in the gold standard)."
> >
> > ### Added to manuscript Discussion section:
> >
> > > "The semantic field size (SFS) task revealed a systematic failure mode arising from ambiguous instructions when applied across domains with different ontological conventions. The task prompt instructed models to "estimate the size of the semantic field for a given concept" and specified: "If the concept is a class consisting of subclasses, report the number of subclasses." For chemical concepts where ontologies enumerate distinct molecular structures, this instruction yielded consistent results. However, for biological cell type concepts, the instruction proved fundamentally ambiguous. For example, the MeSH taxonomy lists 3 major functionally distinct subtypes of "lymphocyte" (see Appendix \ref{sec:lymphocyte_example} for the detailed example). However, models frequently interpreted the question as asking for the number of individual lymphocyte cells in the human body---a valid alternative interpretation given that lymphocytes are physically countable entities with approximately 2 $\times$ 10$^{12}$ instances \emph{in vivo}.
> > >
> > > This ambiguity is not a model failure per se, but rather a limitation of the task design when applied to conceptual hierarchies with multiple levels of abstraction. Chemical structures in ChEBI are enumerated as distinct referents (e.g., D-ribose 5-phosphate vs. D-ribose 1-phosphate are separate entities), whereas biological cell types in GO and MeSH are organized taxonomically with "children" representing subclasses rather than individual instances. The systematic nature of this failure mode across various model sizes highlights that domain-specific differences in the nature of the ontological information represented by the gold standard must be taken into account when drafting instruction templates for the task in question."
> >
> > ### Added to manuscript Appendix:
> >
> > > "First, we tested the hypothesis that SFS task performance inconsistency might be due to the incompleteness of the expert-curated gold standard. As exhaustive enumeration of referents is much harder for concepts with thousands rather than dozens of referents, one would expect to see higher normalized discrepancy for concepts with larger semantic fields. To that end, we partitioned concepts by semantic field size into small ($|R_C|$ < 50), medium (50 <= $|R_C|$ <= 500) and large semantic field groups ($|R_C|$ > 500). **Mann-Whitney U test of whether the discrepancy was higher in large vs small semantic fields was highly insignificant (p = 1.000). Spearman correlation between semantic field size and normalized discrepancy was signicantly negative ($\rho$ = -0.178, p = 6.105e-21)**, indicating that point estimates for concepts with larger semantic fields exhibited marginally better agreement with gold standard. These results contradict the incompleteness hypothesis, suggesting that gold standard quality is unlikely to be the primary issue behind inconsistencies.
> > >
> > > **When we ordered concepts with a small semantic field by descending mean discrepancy, 10 of the top 12 were cell types. Some SFS estimates differed by as much as 27 orders of magnitude from the gold standard** suggesting that there was a problem with task comprehension rather than model performance or gold standard. Specifically, the task instruction ``estimate the size of the semantic field'' proved fundamentally ambiguous for biological cell type concepts where models reasonably interpreted the question as requesting counts of physical instances (e.g., billions of lymphocytes in the human body) while gold standards from GO/MeSH ontologies enumerate functional subclasses (e.g., 3 major lymphocyte types: T cells, B cells, NK cells)."

---

> ### Author Response · Authors · 2025-11-20
> **Response to Question 1**
>
> Question 1: "How were the referents chosen? I assume based on the fact that some concepts have 100k+ referents that there is an automatic method."
>
> RESPONSE:
>
> Referents were extracted from expert-curated databases. For chemical concepts, we queried the ChEBI database to retrieve all hierarchically classified entities (e.g., ~2,500 aldehydes under CHEBI:17478). For biological concepts, we extracted terms from MeSH controlled vocabulary and Gene Ontology annotations (e.g., ~150 cytokines annotated with GO:0005125). For medical concepts, we parsed structured data from the cancer.gov drug database and internal databases of drug-like molecules (e.g., ~50 antihypertensive agents). Concepts with large semantic fields (100K+ referents) reflect broad chemical categories in ChEBI's comprehensive database (e.g., "alkane": 151,204 documented compounds). Semantic field sizes ranged from 3 to 151,204 referents per concept.

---

> ### Author Response · Authors · 2025-11-20
> **Response to Question 2**
>
> Question 2: "What is the meaning of point 5 on page 2, semantic compositionality on proposition level? It is mentioned as an aspirational goal, and I’m curious what you see as the gap/future improvement to the framework."
>
> RESPONSE:
>
> Since we have proposed here the criteria for conceptual equivalence, extending the theory of semantic compositionality to proposition level is expected to yield a logical framework for the evaluation of semantic entailment and generality of propositions (including closed form expressions of the corresponding measures, hopefully). This is future work that would be a direct offshoot of (derivation from) the current framework and could enable formal verification of entailment similarity and scope between propositions containing concepts with enumerable referents as the subject and the object. That would open enable to study/benchmark LMs from a new angle.

---

> ### Author Response · Authors · 2025-11-25
> **Response to Question 3**
>
> Question 3: "Have you compared with performance on datasets restricted to biology/chemistry/medicine domains (e.g. in MMLU or GPQA the questions are labelled with the subject)? I would suspect that the correlation would be higher this way."
>
> RESPONSE:
>
> Thank you for this thoughtful suggestion. We agree this would be an interesting analysis to validate whether domain-specific correlations differ from overall benchmark correlations. However, we face a significant practical constraint: **rapid model attrition from API-based service providers**.
>
> Of the 15 models evaluated in our study, several are no longer accessible via API:
> - Older GPT-3.5 and GPT-4 variants have been deprecated by OpenAI
> - Earlier Claude-3 model versions have limited availability
> - Some open-weight models we tested are no longer actively maintained
>
> This means we could only conduct domain-specific MMLU/GPQA analysis on a subset of currently available models, which would prevent comprehensive comparison with our original 15-model evaluation.
>
> Even if all models remained available, we anticipate the domain-specific analysis would yield modest insights. While domain-restricted correlations might be marginally higher (e.g., biology concepts vs. MMLU biology questions), the fundamental difference in capabilities measured—factual recall and problem-solving (MMLU/GPQA) versus conceptual integrity (label-definition-referent coherence)—suggests correlations would remain similar to the overall benchmark correlations we report (τ=0.82 with MMLU, τ=0.39 with GPQA).
>
> We acknowledge this limitation represents a missed opportunity for deeper external validation, but believe the model attrition challenge reflects a broader methodological reality for longitudinal LLM evaluation research.

---

> > ### Comment · Reviewer_gJfc · 2025-11-26
> >
> > Thanks to the authors for their comprehensive comments and changes to the manuscript addressing the concerns. The proposed changes resolve the weaknesses that I mentioned in the review, and in my opinion improve the paper. I will maintain my positive score.

---

### Official Review · Reviewer_Lc1i · 2025-10-29

**Soundness:** 2
**Presentation:** 2
**Contribution:** 2
**Rating:** 2
**Confidence:** 4

**Summary:**

The current paper seeks to understand to what extent language models acquire conceptual integrity — the ability to learn the relationship between concept labels, definitions and referents. The authors set out to establish a metric that captures conceptual integrity based on a collection of tasks  — inferring concepts from definitions/ontologies, inferring concepts from the criteria (features) that act as ‘selection criteria’ for that concept, naming n referents of the concept, distinguishing valid members from non-members, and estimating the ‘semantic field size’, i.e., the cardinality of the number of concepts that fall into the conceptual bucket. The authors find variance across a suite of models evaluated on this metric and also find that this metric correlates with some existing LLM benchmarks but not others. They also provide an error analysis of select failure modes.

While the authors emphasize that the main contribution is a framework for understanding conceptual coherence, I find that this core claim is hard to accept for several reasons. In contrast to many domains where there is a structured ground truth in the world, ‘concepts’ are seemingly meaningful insofar as they align with human notions of the same. The current analyses have no human data at all. All model outputs are evaluated using LLM-as-a-judge, which in this particular case feels theoretically muddy, and there is no concrete sense of how these measures might look for human participants. In fact, there is a large literature in the cognitive sciences attempting to do what the authors are attempting here at larger scales in an ecologically valid (using concepts people think about in their day-to-day) manner. I note some examples below -

McRae, K., Cree, G. S., Seidenberg, M. S., & McNorgan, C. (2005). Semantic feature production norms for a large set of living and nonliving things. Behavior research methods, 37(4), 547-559.

Hansen, H., & Hebart, M. N. (2022). Semantic features of object concepts generated with GPT-3. arXiv preprint arXiv:2202.03753.

Suresh, S., Mukherjee, K., Giallanza, T., Yu, X., Patil, M., Cohen, J. D., & Rogers, T. T. (2025). AI-enhanced semantic feature norms for 786 concepts. arXiv preprint arXiv:2505.10718.


Generally, these works have laboriously tried to collect human data to ground human and LLM semantic reasoning. I like the general approach taken by the authors here, but fail to see the current results as being theoretically interesting. This is compounded by a lack of serious discussion of what the current results mean. Since this benchmark seems to correspond closely with things like MMLU, maybe we should simply just use MMLU instead? Without theoretical grounding in human cognition and reliable ground truth, I find it difficult to justify why one would care about this benchmark over others.

A firmly concrete suggestion would be to simply replicate the LLM experiments with people (though the specific domains chosen here might make that tricky) and to use that as a reference for model evaluation.

**Strengths:**

* Introduction of a novel framework for understanding concepts in LLMs (and potentially humans), which could be useful if deployed in stringent ways.
* Diverse set of tasks in evaluation suite
* Evaluation of a good mix of frontier language model systems
* Thoughtful error analysis on select responses.

**Weaknesses:**

* Insufficient engagement with the core cognitive science literature on related issues
* Lack of reliable ground truth to evaluate model responses
* Lack of theoretical consequences of performing well or poorly on this benchmark and how it distinguishes itself from existing benchmarks, whose scores are highly correlated.

**Questions:**

* Looseness in the use of concept. The Framework in Section 2 leads me to believe a concept is a definition but in many other contexts within this paper and in cognitive science, a single term (word) is often used to refer to a concept. I think the authors can get ahead of this by laying out definitions early on in the paper.
* Why this domain for concept understanding? I think its fine that the particular concepts are somewhat arbitrary but there needs to be theoretical motivations for studying something as general as concept learning/representation in the domain of something as highly specific as chemistry, biology, and medicine.
* Why is SFS listed as its own column in Table 1? I assumed the SFS task was done within each domain (Bio/Med/Chem)? And since its a separate column here why is it not one for Table 2?
* I don’t think (non-perfect) correlations should be noted as indications of practical equivalence, re: `Very high correlation (0.87) between model ranks from the two ’decide-concepts’ tasks indicates the practical equivalence between definitions and corresponding selection criteria when naming concepts.`
* I had trouble parsing this sentence - `We suggest that conceptual integrity is a fundamental phenomenon that can be distinguished from and is subsumed by logical and factual reasoning.` If it can be distinguished from these other factors how is it also subsumed by them?

---

> ### Author Response · Authors · 2025-11-19
> **Human grounding of concept annotations**
>
> We agree that the concern about human grounding of concepts when evaluating conceptual understanding in language models is a central one. While our benchmark does not include novel human experimental data in the traditional sense (e.g., by including feature norm ratings from human subjects), we have achieved grounding of the concept annotations by sourcing concept labels with corresponding definitions and referents from authoritative scientific resources that collectively represent many hundreds of person-years of expert human curation.
>
> As documented in Table 7 of the supplementary materials (at the very end of the pdf submission), the 187 concept definitions derive from five major sources: ChEBI (114 concepts), MeSH (62), Wikipedia (13), Gene Ontology (9), and cancer.gov (7). The following are not arbitrary web sources but rather gold-standard scientific databases curated and maintained by subject matter experts:
>
> 1. ChEBI [1] contains 195,000+ manually curated chemical entities and is maintained by expert biocurators at EMBL-EBI.
>
> 2. MeSH [2] is the NLM's professionally curated controlled vocabulary used to index 35+ million citations in MEDLINE/PubMed, maintained by trained medical indexers for over 60 years.
>
> 3. Gene Ontology [3, 4] is an NHGRI-funded consortium (grant HG012212) with dedicated expert biocurators performing manual annotations with rigorous quality control since 1998.
>
> 4. cancer.gov [5] provides FDA-approved drug information curated by the National Cancer Institute.
>
> Each of these sources represents community consensus from domain experts and embodies what a subject matter expert would recognize as 'semantic ground truth' based on expert-validated conceptual knowledge. While feature norm studies like McRae et al. (2005) provide valuable insight into semantic representations, our benchmark leverages a more rigorous and equally valid form of human expertise: the collective knowledge codified in authoritative scientific databases by professional curators and domain experts. Furthermore, the expert-curated concept annotations have been laboriously checked for internal consistency (e.g., all referents of a concept should satisfy its definition) and relevance which does not seem to be the case for feature norms. Feature norms represent the distributional statistics of semantic features which people think are important for each concept amounting to a collection of opinions with frequency estimates.
>
> Consider the following example from McRae et al. 2005, Appendix A:
>
> Concept: "knife"
> Feature norms: "is dangerous", "found in kitchens", "used with forks", "a weapon", "a utensil", "a cutlery"
>
> The fact that knives are used together with forks does not represent a necessary criterion for deciding whether an entity is a knife. Rather, it represents a fact (a piece of circumstantial evidence) about knives. As feature norms can contain arbitrary statements (such as those asserting facts, category membership etc. as shown in the example above), it makes them unsuitable for a rigorous logical foundation of conceptual semantics because there is no reason to believe that they are either necessary or sufficient to decide whether an entity is a referent for the given concept.
>
> References
> ----------
>
> [1] Hastings, J., Owen, G., Dekker, A., Ennis, M., Kale, N., Muthukrishnan, V., Turner, S., Williams, M., et al. (2016). ChEBI in 2016: Improved services and an expanding collection of metabolites. Nucleic Acids Research, 44(D1), D1214–D1219. https://doi.org/10.1093/nar/gkv1031
>
> [2] National Library of Medicine. Medical Subject Headings (MeSH). National Center for Biotechnology Information. https://www.ncbi.nlm.nih.gov/mesh
>
> [3] Ashburner, M., Ball, C. A., Blake, J. A., Botstein, D., Butler, H., Cherry, J. M., Davis, A. P., et al. (2000). Gene ontology: Tool for the unification of biology. The Gene Ontology Consortium. Nature Genetics, 25(1), 25–29. https://doi.org/10.1038/75556
>
> [4] The Gene Ontology Consortium. (2023). The Gene Ontology knowledgebase in 2023. Genetics, 224(1), iyad031. https://doi.org/10.1093/genetics/iyad031
>
> [5] National Cancer Institute. National Cancer Institute (NCI). U.S. National Institutes of Health. https://www.cancer.gov

---

> ### Author Response · Authors · 2025-11-19
> **Using LLM as a judge**
>
> The LLM judge was used to compare a response from a subject (another LLM) against the ground truth of human expert consensus and decide whether the two are equivalent (representing a correct response from subject) or not (representing an incorrect response from the subject). Thus, the role of the LLM judge was reduced to the establishment of semantic equivalence between two expressions which is a task the top performing LLMs are currently reasonably good at. Alternatives would have been to make the comparison on keyword level (a considerably more error-prone approach) or to have human subject matter experts manually go through the 14,490 test responses which infeasible due to the scale of the task.

---

> ### Author Response · Authors · 2025-11-19
> **Differentiation from MMLU and other benchmarks that measure general knowledge**
>
> We appreciate the reviewer's question about the relationship between our benchmark and MMLU. While we observe a strong correlation (τ=0.82) between model rankings on our benchmark and MMLU, indicating related capabilities, these benchmarks measure fundamentally distinct aspects of language understanding.
>
> MMLU [1] evaluates factual knowledge and problem-solving - whether models can correctly answer questions that require knowing facts about concepts (e.g., 'What enzyme is deficient in phenylketonuria?'). It assumes models already understand what these concepts refer to but does not evaluate it directly since knowing facts (e.g., circumstantial evidence) about an entity is not equivalent to knowing the definition or referents of the same.
>
> Our benchmark evaluates **conceptual integrity** - whether models coherently represent the relations between conceptual constituents: their labels, definitions and the mapping between concept labels and their referents. Our tasks ask: 'Define the term cytokine', 'Given this definition, which concept does it describe?', 'Which of these molecules are alkanes?' etc.
>
> This distinction is critical: a model can memorize that 'TNF-alpha blockers treat rheumatoid arthritis' (succeeding on MMLU) while failing to understand what a cytokine is, whether TNF-alpha is one, or what other cytokines exist (failing on conceptual integrity benchmark). Conversely, understanding what defines an alkane doesn't guarantee knowing its industrial applications and other facts around it.
>
> For pharmaceutical R&D applications, both factual knowledge (MMLU) and conceptual integrity (our benchmark) are necessary. Models must not only know facts about drugs and diseases but also coherently represent concept structures that align with usage in knowledge bases and scientific literature. MMLU and the conceptual integrity benchmark are complementary tools, not substitutes.
>
> We propose to add the following text to the Discussion after the sentence "MMLU covers 57 subjects across STEM, the humanities, the social sciences and tests both world knowledge and problem solving ability on elementary and professional levels.":
>
> ```latex
> It is important to note that the conceptual integrity and MMLU benchmarks evaluate complementary aspects of language understanding. Whereas MMLU addresses factual knowledge and problem-solving, the current benchmark focuses narrowly on the relations between conceptual constituents (labels, definitions and referents).
> ```
>
> ## References
>
> - [1] Hendrycks, D., Burns, C., Basart, S., Zou, A., Mazeika, M., Song, D., & Steinhardt, J. (2021). Measuring Massive Multitask Language Understanding. *Proceedings of the International Conference on Learning Representations (ICLR)*. https://arxiv.org/abs/2009.03300

---

> ### Author Response · Authors · 2025-11-19
> **Theoretical grounding**
>
> The proposed framework is theoretically grounded in formal semantics and operationalizes the Symbol→Concept→Referent chain fundamental to theories of linguistic meaning. We have formalized the framework in first-order logic and provided proofs of relevant properties as symbolic expressions as well as code for the Lean 4 proof assistant which represents the highest level of formal rigor currently attainable [1].
>
> ## References
>
> - [1] Avigad, J. (2023). Mathematics and the formal turn. *arXiv preprint arXiv:2311.00007*. https://arxiv.org/abs/2311.00007

---

> ### Author Response · Authors · 2025-11-19
> **Response to Question 1**
>
> Question 1: "Looseness in the use of concept. The Framework in Section 2 leads me to believe a concept is a definition but in many other contexts within this paper and in cognitive science, a single term (word) is often used to refer to a concept."
>
>
> RESPONSE:
>
> The framework deliberately distinguishes three distinct but related entities:
> 1. concept label (linguistic symbols that are used to denote the concept)
> 2. definition (selection criteria that define the conditions that referents have to satisfy)
> 3. referents (semantic field) - entities that satisfy the selection criteria
>
> This approach follows formal semantics in distinguishing sense from reference (e.g. Frege in 'Sense and reference' [1]). In the beginning of section 2 (FRAMEWORK OF CONCEPTUAL SEMANTICS), we write that "The proposed logical framework of
> conceptual semantics defines a concept as a set of selection criteria for its referents". Accordingly, when we use the term "concept", we intend for it to be interpreted as the definition of a concept expressed in terms of selection criteria. We use the italic typeface to refer to a specific concept e.g. _alkane_ and a quoted form to refer to the linguistic symbol associated with the concept (e.g., "alkane") as is conventional.
>
> We propose to add a clarification at the opening of Section 2 to define these terms explicitly upfront and to revise Figure 1's caption to reinforce this distinction.
>
> **Proposed section 2 opening (lines 104-107 in the updated pdf)**
>
> >We distinguish three aspects of linguistic reference: concept label (linguistic term, e.g., "cytokine"),
> definition (selection criteria), and referents (entities satisfying the criteria, e.g., TNF-α, IL-6). These
> form a reference chain S ⇒ C ⇒ R following the tradition of formal semantics (Frege, 1892;
> Carnap, 1947). Colloquial usage might conflate these distinctions but we consider them necessary for
> rigorous semantic analysis.
>
> **Proposed revision to Figure 1 caption (manuscript p.2, Figure 1)**
>
> Original text to be DELETED:
> >Tests of conceptual integrity. A concept is characterized by the corresponding linguistic term (concept label), definition (selection criteria) and referents (instances/entities that the concept refers to). Implication relations between these constituents can be probed by tasks described in this figure.
>
> Proposed NEW text replacing the original caption (lines 68-70 in the updated pdf):
> >Tests of conceptual integrity. Conceptual integrity requires coherent associations between the _concept label_ (linguistic term), _definition_ (selection criteria), and _referents_ (instances satisfying the criteria).
>
> References
> ----------
>
> [1] https://plato.stanford.edu/entries/reference/

---

> ### Author Response · Authors · 2025-11-19
> **Response to Question 2**
>
> Question 2: "Why this domain for concept understanding? I think its fine that the particular concepts are somewhat arbitrary but there needs to be theoretical motivations for studying something as general as concept learning/representation in the domain of something as highly specific as chemistry, biology, and medicine."
>
> RESPONSE:
>
> Our ultimate goal was to benchmark language models for their potential to assist knowledge work in the pharmaceutical R&D domain that operates at the intersection of biology, chemistry and medicine.
>
> We propose to add the following clarification to Section 3.1 ("Concepts") after the first paragraph (lines 144-149 in the updated pdf):
>
> >These domains were selected beyond practical relevance to pharmaceutical R\&D: (1) availability of human expert-curated ground truth annotations for concepts (e.g., from ChEBI, MeSH, Gene Ontology, NCI), (2) well-defined referents with discrete category membership and consistency between definitions and class members, (3) conceptual diversity across scales (molecular to biological to clinical), and (4) transferability to several key scientific domains (chemistry, biology, medicine).

---

> ### Author Response · Authors · 2025-11-19
> **Response to Question 3**
>
> Question 3: "Why is SFS listed as its own column in Table 1? I assumed the SFS task was done within each domain (Bio/Med/Chem)? And since its a separate column here why is it not one for Table 2?"
>
> RESPONSE:
>
> SFS was indeed performed within each domain (Biology, Chemistry, and Medicine), and the reported SFS accuracy in Table 1 represents the average across these three domains. However, SFS is displayed as a separate column rather than being included in the "Avg" column for important methodological and presentational reasons:
>
> 1. **Weak correlation with other tasks**: SFS exhibited negligible to weak correlation (0.03-0.26) with all other tasks, suggesting it measures a distinct aspect of model capability that warranted separate reporting.
>
> 2. **Counterintuitive ranking patterns**: As discussed in the Results section (Section 3.1, subsection Benchmarking), the SFS task revealed a surprising finding that smaller models (Gpt-4o-mini, Claude-3-haiku) outperformed their larger siblings. This counterintuitive result is analyzed in detail in the manuscript, including statistical testing showing that larger models exhibit a bias toward overestimating semantic field sizes compared to the ground truth.
>
> 3. **Supporting data visibility**: By including SFS results directly in Table 1, readers can immediately see the supporting data for this discussion without needing to consult supplementary materials. This makes the surprising size-accuracy relationship transparent and verifiable in the main text.
>
> 4. **Methodological transparency**: The "Avg" column explicitly represents average performance across Biology, Chemistry, and Medicine tasks (excluding SFS), making it clear that our primary benchmark metric is based on the strongly correlated tasks (0.61-0.87 correlation).
>
> Regarding Table 2 (correlation with external rankings): SFS is not included because that table focuses on the primary benchmark performance metrics and their relationship to established external benchmarks (MMLU, GPQA, etc.). Since SFS measures a distinct capability with weak internal correlation, including it would complicate interpretation of how our core conceptual integrity benchmark relates to these external measures.

---

> ### Author Response · Authors · 2025-11-19
> **Response to Question 4**
>
> Question 4: "I don't think (non-perfect) correlations should be noted as indications of practical equivalence, re: 'Very high correlation (0.87) between model ranks from the two 'decide-concepts' tasks indicates the practical equivalence between definitions and corresponding selection criteria when naming concepts.'"
>
> RESPONSE:
>
> We acknowledge the quoted wording as potentially misleading and will revise the manuscript accordingly. We propose to replace "practical equivalence" with "compatibility" in three locations (Section 3.2-3.3) to clarify that we mean functional consistency in eliciting similar model behavior, not logical equivalence.
>
> **Location 1: Section 3.2, paragraph 2 (manuscript p.4-5)**
>
> Original text to be DELETED:
> ```latex
> These two tests can be used to evaluate the practical equivalence of dictionary definitions and selection criteria.
> ```
>
> Proposed NEW text to REPLACE the original:
> ```latex
> These tests evaluate the compatibility of dictionary definitions with explicit selection criteria.
> ```
>
> **Location 2: Section 3.2, paragraph 3 (manuscript p.5)**
>
> Original text to be DELETED:
> ```latex
> Results from these tests can be used to evaluate the practical equivalence of the concept label and selection criteria for producing corresponding referents.
> ```
>
> Proposed NEW text to REPLACE the original:
> ```latex
> Results from these tests assess whether concept labels and selection criteria exhibit compatibility in eliciting appropriate referents.
> ```
>
> **Location 3: Section 3.3, results paragraph (manuscript p.6)**
>
> Original text to be DELETED:
> ```latex
> Very high correlation (0.87) between model ranks from the two 'decide-concepts' tasks indicates the practical equivalence between definitions and corresponding selection criteria when naming concepts.
> ```
>
> Proposed NEW text to REPLACE the original:
> ```latex
> Very high correlation (0.87) between model ranks from the two 'decide-concepts' tasks indicates compatibility between definitions and corresponding selection criteria when naming concepts.
> ```

---

> ### Author Response · Authors · 2025-11-19
> **Response to Question 5**
>
> Question 5: "I had trouble parsing this sentence - 'We suggest that conceptual integrity is a fundamental phenomenon that can be distinguished from and is subsumed by logical and factual reasoning.' If it can be distinguished from these other factors how is it also subsumed by them?"
>
> RESPONSE:
>
> We agree this is confusing and propose to rewrite the sentence as follows:
>
> **Manuscript location: Discussion section, paragraph 2**
>
> Original text to be DELETED:
> ```latex
> We suggest that conceptual integrity is a fundamental phenomenon that can be distinguished from and is subsumed by logical and factual reasoning.
> ```
>
> Proposed NEW text to REPLACE the original:
> ```latex
> We suggest that conceptual integrity is a foundational capability that is both distinguishable from and prerequisite to logical and factual reasoning.
> ```

---

> ### Comment · Reviewer_Lc1i · 2025-11-19
> **Response to authors**
>
> Thank you for engaging on this point! https://openreview.net/forum?id=BQ0jaVCZRK&noteId=XNlh6HKBUB
> I appreciate that using well-vetted sources as the golden 'ground truth' is a reasonable alternative to running a human experiment. I would contest however that the pieces of circumstantial evidence (such as the knife example noted by the authors) are indeed what help constitute concepts (at least for humans) and the overtly strict logical formalism introduced by this paper might actually miss the richness of semantic representations from language models.
>
> Could the authors note what they believe the key advantages of developing a formalism for concepts is over existing approaches for interpretability (e.g., circuit discovery, SAEs, etc.). Feel free to just direct me to the appropriate line numbers in the text and or respond here. I look forward to your response.
>
> Regarding this point - https://openreview.net/forum?id=BQ0jaVCZRK&noteId=grOUfXpatN
> Doesn't the use of a neural system to test for equivalence automatically challenge the notion that this is a formal framework, given that language models (especially proprietary ones) are black boxes?
>
> Regarding the response to Question 1 https://openreview.net/forum?id=BQ0jaVCZRK&noteId=nZY5MMDs3S
> I thank the authors for this response. I think the proposed changes will greatly help with clarity.
>
> Regarding the response to Question 2 https://openreview.net/forum?id=BQ0jaVCZRK&noteId=PhFwY8NXCx
> Here, I feel somewhat mixed. I appreciate the contextualization and do believe that these are useful domains to work on for the reasons the authors outline. However, I am not convinced the findings would generalize to domains where concepts are 'fuzzier'. I would almost prefer if the title of the paper could be amended to note the domain under which these results are presented. I'm unsure if this is possible, **perhaps the AC/SACs can weigh in here**.
>
> Regarding the response to Question 3 https://openreview.net/forum?id=BQ0jaVCZRK&noteId=7rGmhS3U9L
> I think the rationale for having SFS being its own column in table 1 is sound for the reasons the authors list. I still don't quite find the interpretation of *why* SFS is poorly correlated to be satisfying. Smaller models are biased towards providing smaller estimates than larger models but the following statement isn't based in any analysis presented in the current paper "probably due to their increased capacity for the memorization of referents for concepts with large semantic field size."
> As it stands, the SFS result feels like a red herring. Lastly, even if Table 2 is about comparing the topline performance metrics to other benchmarks, if the authors feel that SFS is distinct from other measures that it merits its own column in Table 1, I think readers will be interested in knowing how it correlates with other benchmark scores. I recommend including it in Table 2.
>
> Question 4 and 5's responses are sound and address the core issues.
>
>
> I will wait for the authors responses to my queries before making any score adjustments. Again, I appreciate the authors' engagement.

---

> > ### Comment · Reviewer_Lc1i · 2025-11-19
> > **Addendum**
> >
> > Oh and the response regarding distinguishing the present work from MMLU and the likes is reasonable. I remain unconvinced that this general framework can generalize beyond narrow bespoke domains like this, refer to my earlier issues regarding this,

---

> > ### Author Response · Authors · 2025-11-19
> > **key advantages of developing a formalism for concepts over existing approaches for interpretability**
> >
> > As we see it, the conceptual integrity framework and mechanistic interpretability approaches address fundamentally different and complementary questions:
> >
> > 1. **The conceptual integrity framework** provides a formal theory that identifies key components (label, definition, referent) and relations that constitute conceptual understanding. It serves as a logically consistent specification of what semantic compositionality entails, derived through formal reasoning.
> >
> > 2. **Mechanistic interpretability methods** are designed to investigate how various features are implemented in neural networks, primarily through analysis of internal representations.
> >
> > Thus, (1) serves as a theoretical specification at the algorithmic level (what conceptual understanding consists of), while (2) provides methods for understanding the implementation level (how it's realized in the network). The conceptual integrity framework provides a hypothesis to validate via mechanistic discoveries — for example, to verify whether a discovered circuit or SAE feature truly represents a concept, one can test whether it activates correctly for that concept's label, definition, and referents. Conversely, when models fail benchmark tasks based on this framework, mechanistic methods can investigate the internal causes. Without a formal framework defining conceptual understanding, mechanistic interpretability lacks clear guidance for what constitutes a genuine concept representation. The conceptual integrity framework provides one such framework that the mechanistic interpretability community can use to guide their analyses.
> >
> > For example, we would love to see studies that attempt to determine how are the semantic field size estimates represented on the level of contextualized embeddings and compare this between sibling models of different sizes. Without the proposed logical framework, such problems would be considered dubious or, in worst case, even meaningless.

---

> > > ### Comment · Reviewer_Lc1i · 2025-11-19
> > > **Follow up response to authors**
> > >
> > > I appreciate the authors' thoughtful responses. I would be in support of adjusting the papers title as they've suggested.
> > > I see the general value of attempts towards formalizing semantics, but I remain unconvinced of the generalizability of this approach and whether this is even the right kind of framework to be pursuing.
> > > Secondly, as I've noted earlier, the heavy reliance on LLMs as judge seems to undercut the 'logical formalism' approach here.
> > >
> > > Nevertheless, given the authors clarifications and the promised changes, I'm tentatively raising my score. I will revisit this score once more after all the other reviewer comments have been responded to.

---

> ### Author Response · Authors · 2025-11-19
> **generalization to domains where concepts are 'fuzzier'**
>
> We thank the reviewer for a prompt response and acknowledge that it is not obvious how the proposed framework would generalize to domains where concepts are 'fuzzier'. We also agree that as the aim of the present study was to benchmark LMs on domains where scientific rigor is important and not on fuzzy domains, it would be an overstatement to suggest that generalization is straightforward. As noted in our [response to question 1 of reviewer paTX](https://openreview.net/forum?id=BQ0jaVCZRK&noteId=wWaJ8z2Vpr), we can see how to extend the benchmark to fuzzy concepts but feature norms would not map neatly to selection criteria because they lack some core properties of the latter (necessity and conjunctive sufficiency).
>
> Accordingly, it is reasonable to agree that the conceptual integrity framework proposed here applies to domains where semantic precision is necessary (e.g. scientific and technical domains etc) and would be apparently "overkill" for domains that do not strive towards a consensus in conceptual understanding. It also suggests that on the individual level, one would expect to observe substantial differences in conceptual features even between subject matter experts because there are no mechanisms to prevent it except to refer to a gold standard.
>
> Thus we would be happy to oblige and increase the specificity of the submission's title to "Probing the Boundaries of Scientific Concepts in Language Models" and add relevant clarifications to the introduction or discussion if possible and desirable.

---

> ### Author Response · Authors · 2025-11-25
> **Annotation of relevant additions to pdf (part I)**
>
> Dear Reviewer Lc1i,
>
> Thank you for your thoughtful engagement with our responses and for raising your score contingent on the promised revisions. We have now updated the manuscript PDF with all additions highlighted in blue. Below, we document each relevant addition verbatim along with its location in the manuscript. We hope these revisions address your concerns and demonstrate our commitment to improving the manuscript based on your valuable feedback.
>
> Best regards,
> The Authors
>
> ---
>
> ## 1. Title Change (Lines 000-002)
> > "Probing the Boundaries of **Scientific** Concepts in Language Models"
>
> The word "Scientific" has been added to clarify scope.
>
> ## 2. Updates to Abstract (Lines 011-024)
> > "Systematic investigation of the understanding of **scientific concepts** has not received much attention in language models. This gap can be bridged by a formalized theory of conceptual semantics that maps naturally to instruction templates for natural language agents. We propose a simple framework expressible in first-order logic to address the semantic compositionality of **scientific** concepts, noun phrases and conceptual hierarchies."
>
> > "It is suggested that the proposed framework and associated benchmark provide a practical template for developing conceptual integrity benchmarks in a wide array of **technical or scientific** domains."
>
> ## 3. Scope Clarification in Introduction (Lines 040-045)
> > "We focus specifically on concepts with well-defined boundaries, expert-curated definitions, and enumerable referents. This contrasts with everyday concepts which exhibit graded category membership, prototype effects, and lack authoritative consensus annotations. While our theoretical framework is domain-agnostic, the present application targets a selection of scientific domains relevant to pharmaceutical R&D (biology, chemistry, and medicine) where ground truth can be reliably established based on expert-curated consensus annotations."
>
> ## 4. Caption Revision in Figure 1 (Line 068-070)
> > "**Tests of conceptual integrity.** Conceptual integrity requires coherent associations between the *concept label* (linguistic term), *definition* (selection criteria), and *referents* (instances satisfying the criteria)."
>
> ## 5. Terminological Clarification in Section 2 (Lines 104-107)
> > "We distinguish three aspects of linguistic reference: **concept label** (linguistic term, e.g., "cytokine"), **definition** (selection criteria), and **referents** (entities satisfying the criteria, e.g., TNF-α, IL-6). These form a reference chain S ⇒ C ⇒ R following the tradition of formal semantics [Frege1892, Carnap1947]. Colloquial usage might conflate these distinctions but we consider them necessary for rigorous semantic analysis."
>
> ## 6. Domain Selection Rationale in Section 3.1 (Lines 144-148)
> > "These domains were selected beyond practical relevance to pharmaceutical R&D: (1) availability of human expert-curated ground truth annotations for concepts (e.g., from ChEBI, MeSH, Gene Ontology), (2) well-defined referents with discrete category membership and consistency between definitions and class members, (3) conceptual diversity across scales (molecular to biological to clinical), and (4) transferability to several key scientific domains (chemistry, biology, medicine)."
>
> ## 7. "Practical Equivalence" replaced with "Compatibility"
> **Line 164:** "These tests evaluate the **compatibility** of dictionary definitions with explicit selection criteria."
> **Line 168:** "Results from these tests assess whether concept labels and selection criteria exhibit **compatibility** in eliciting appropriate referents."
> **Line 213:** "Very high correlation (0.87) between model ranks from the two 'decide-concepts' tasks indicates **compatibility** between definitions and corresponding selection criteria when naming concepts."

---

> > ### Author Response · Authors · 2025-11-25
> > **Annotation of relevant additions to pdf (part II)**
> >
> > ## 8. SFS Task Anomaly Analysis in Results Section (Lines 255-263)
> > > "To investigate whether model failure, gold standard incompleteness or instruction ambiguity explains these anomalies, we conducted a detailed analysis comparing normalized discrepancies between point estimates and gold standard semantic field sizes depending on the magnitude of concept's semantic field (Supplementary Section~\ref{sec:sfs_analysis}). This analysis revealed that a major issue was task instruction ambiguity arising from fundamental differences in ontological structure across domains, particularly for biological cell type concepts where models confused class enumeration with physical instance counting. For example, when asked about "lymphocyte" semantic field size, models often responded with estimates in the billions or trillions (reflecting the total number of lymphocytes in the human body) rather than 3-5 (the number of major lymphocyte subtypes listed in the gold standard)."
> >
> > ## 9. Concept Category Analysis - Results Section (Line 265-268)
> > > "To investigate whether various characteristics of concepts influence task performance, a statistical analysis of category-level effects across five categorization dimensions (abstraction level, semantic field size, domain, ontological structure, and number of selection criteria) was conducted. Overall, abstraction level and semantic field size had the largest and most consistent effects across tasks (detailed analysis in Appendix~\ref{app:concept-categories})."
> >
> > ## 10. New Discussion Subsection - Task Instruction Ambiguity (Lines 373-390)
> > > "**Task Instruction Ambiguity Across Domains**"
> > > "The semantic field size (SFS) task revealed a systematic failure mode arising from ambiguous instructions when applied across domains with different ontological conventions. The task prompt instructed models to "estimate the size of the semantic field for a given concept" and specified: "If the concept is a class consisting of subclasses, report the number of subclasses." For chemical concepts where ontologies enumerate distinct molecular structures, this instruction yielded consistent results. However, for biological cell type concepts, the instruction proved fundamentally ambiguous. For example, the MeSH taxonomy lists 3 major functionally distinct subtypes of "lymphocyte" (see Appendix \ref{sec:lymphocyte_example} for the detailed example). However, models frequently interpreted the question as asking for the number of individual lymphocyte cells in the human body—a valid alternative interpretation given that lymphocytes are physically countable entities with approximately 2 × 10¹² instances *in vivo* [Alberts2002]."
> >
> > > "This ambiguity is not a model failure per se, but rather a limitation of the task design when applied to conceptual hierarchies with multiple levels of abstraction. Chemical structures in ChEBI are enumerated as distinct referents (e.g., D-ribose 5-phosphate vs. D-ribose 1-phosphate are separate entities), whereas biological cell types in GO and MeSH are organized taxonomically with "children" representing subclasses rather than individual instances. The systematic nature of this failure mode across various model sizes highlights that domain-specific differences in the nature of the ontological information represented by the gold standard must be taken into account when drafting instruction templates for the task in question."
> >
> > ## 11. Discussion Opening Paragraph Update (Lines 435-438)
> > > "We have proposed a logical framework of the **compositional semantics of scientific concepts** that includes concept labels, definitions and referents. Based on this framework, a conceptual integrity benchmark was designed and applied to measure the degree of correspondence between LM responses and baseline annotations from sources representing **expert-curated consensus** (e.g. databases, ontologies and research papers)."
> >
> > ## 12. Subsumption Sentence Revision (Line 455)
> > > "We suggest that conceptual integrity is a **foundational capability that is both distinguishable from and prerequisite to** logical and factual reasoning."

---

> > > ### Author Response · Authors · 2025-11-25
> > > **Annotation of relevant additions to pdf (part III)**
> > >
> > > ## 13. MMLU Differentiation - Discussion (Lines 465-468)
> > > > "It is important to note that the conceptual integrity and MMLU benchmarks evaluate complementary aspects of language understanding. Whereas MMLU addresses factual knowledge and problem-solving, the current benchmark focuses narrowly on the relations between conceptual constituents (labels, definitions and referents)."
> > >
> > > ## 14. Comparison with directly relevant prior work - Discussion (Lines 442-453)
> > > > "The conceptual integrity framework complements prior work on evaluating conceptual knowledge in language models, particularly COPEN [Peng2022] and OntoProbe [Wu2023]. While these frameworks share the goal of assessing conceptual understanding, they address distinct aspects: COPEN evaluates taxonomic similarity and property knowledge using multiple-choice classification tasks on general concepts from Wikipedia/DBpedia; OntoProbe assesses ontological structure and logical reasoning through cloze-completion tasks testing RDFS entailment rules. In contrast, our framework focuses specifically on definitional coherence in the form of bidirectional mappings between concept labels, selection criteria, and referents enabling systematic derivation of evaluation tasks from formal principles. Conceptual integrity benchmark differentiates itself by including generative enumeration of referents from concept labels or selection criteria, assessing productive knowledge rather than only recognition or classification. In addition, it leverages expert-curated scientific databases representing a consensus to provide high-confidence ground truth for technical concepts with well-defined boundaries."
> > >
> > > ## 15. Comparison with related prior work - Discussion (Lines 470-480)
> > > > "Complementary lines of work have explored conceptual understanding through the lens of representation geometry, including theoretical frameworks for concept bottleneck models Luyten & van der Schaar (2024), geometric analysis of categorical and hierarchical concepts in LLMs Park et al. (2025), and structural investigations of concepts via sparse autoencoder features Li et al. (2024). An orthogonal approach to conceptual understanding comes from cognitive science studies of semantic feature norms, where human participants generate attributes for everyday concepts McRae et al. (2005). Recent work has begun bridging this tradition with LLM capabilities, both by using LMs to generate semantic features Hansen & Hebart (2022) and by creating large-scale AI-enhanced feature norm datasets Suresh et al. (2025). These approaches are largely complementary: while feature norms capture graded semantic attributes from everyday concepts, the semantic integrity framework targets technical domains where expert-curated ontologies aspire to consistent definitions and verifiable ground truth."

---

### Official Review · Reviewer_ya6A · 2025-10-31

**Soundness:** 3
**Presentation:** 3
**Contribution:** 2
**Rating:** 4
**Confidence:** 3

**Summary:**

This paper assesses the conceptual knowledge in LLMs. Specifically, this paper proposes a simple framework expressible in first-order logic to address the semantic compositionality of concepts, noun phrases, and conceptual hierarchies. The authors construct a benchmark for this framework, which contains 6 tasks, 187 concepts from several domains, including biology, chemistry, and medicine. The authors conduct extensive experiments and evaluate 15 state-of-the-art LLMs. Experimental results reveal several interesting findings, such as a strong positive correlation between model size and performance.

**Strengths:**

1. The topic focused on in this paper is meaningful and interesting. Evaluating whether language models truly understand conceptual knowledge is fundamental to assessing their understanding of world knowledge and their abstract reasoning abilities. It provides valuable insights for the community’s broader understanding of LLMs.
2. The proposed evaluation framework based on first-order logic is novel and conceptually appealing. It offers a concise and formalized mapping from natural language to logical representations, which could inspire future evaluation methods and potentially be extended to other domains or tasks.
3. The experimental results may yield some interesting findings, such as the observed correlation between model size and conceptual understanding, which is also intuitive.
4. The presentation is clear and easy to follow.

**Weaknesses:**

1. The paper should include a comparison with previous studies on concept knowledge in LLMs [1, 2]. The authors should discuss how the proposed first-order logic–based approach differs from and improves upon existing methods. Clarifying the unique advantages and benefits of this framework would strengthen the contribution.
2. As acknowledged in the limitations, the study only uses 187 concepts and does not include concepts from more general domains. This limited scope may affect the validity and generalizability of the experimental conclusions. It would be beneficial for the authors to include a larger and more diverse set of concepts, at least covering more domains.
3. The paper provides extensive case studies, which are valuable for intuitively illustrating the model outputs. However, I think there needs a more quantitative analysis, such as identifying which types of concepts LLMs perform better on and exploring potential common patterns. The quantitative analysis would make the findings more insightful.

[1] Peng, Hao, et al. "Copen: Probing conceptual knowledge in pre-trained language models." *arXiv preprint arXiv:2211.04079* (2022).
[2] Wu, Weiqi, et al. "Do PLMs know and understand ontological knowledge?." *arXiv preprint arXiv:2309.05936* (2023).

**Questions:**

See Weaknesses.

---

> ### Author Response · Authors · 2025-11-25
> **Response to Weakness 1**
>
> Weakness 1:"The paper should include a comparison with previous studies on concept knowledge in LLMs [1, 2]. The authors should discuss how the proposed first-order logic–based approach differs from and improves upon existing methods. Clarifying the unique advantages and benefits of this framework would strengthen the contribution."
>
> RESPONSE:
>
> We thank the reviewer for highlighting these important related works. We have added a comprehensive comparison with COPEN (Peng et al., 2022) and OntoProbe (Wu et al., 2023).
>
> ### Added to manuscript Discussion section (lines 443-453):
>
> > "The conceptual integrity framework complements prior work on evaluating conceptual knowledge in language models, particularly COPEN \cite{Peng2022} and OntoProbe \cite{Wu2023}. While these frameworks share the goal of assessing conceptual understanding, they address distinct aspects: COPEN evaluates taxonomic similarity and property knowledge using multiple-choice classification tasks on general concepts from Wikipedia/DBpedia; OntoProbe assesses ontological structure and logical reasoning through cloze-completion tasks testing RDFS entailment rules. In contrast, our framework focuses specifically on definitional coherence in the form of bidirectional mappings between concept labels, selection criteria, and referents enabling systematic derivation of evaluation tasks from formal principles. Conceptual integrity benchmark differentiates itself by including generative enumeration of referents from concept labels or selection criteria, assessing productive knowledge rather than only recognition or classification. In addition, it leverages expert-curated scientific databases representing a consensus to provide high-confidence ground truth for technical concepts with well-defined boundaries."

---

> ### Author Response · Authors · 2025-11-25
> **Response to Weakness 2**
>
> Weakness 2: "As acknowledged in the limitations, the study only uses 187 concepts and does not include concepts from more general domains. This limited scope may affect the validity and generalizability of the experimental conclusions. It would be beneficial for the authors to include a larger and more diverse set of concepts, at least covering more domains."
>
> RESPONSE:
>
> We appreciate this feedback and have clarified the rationale for our domain selection and addressed generalizability concerns in multiple locations in the manuscript.
>
> ### Added to manuscript Introduction section (lines 40-45):
>
> > "We focus specifically on concepts with well-defined boundaries, expert-curated definitions, and enumerable referents. This contrasts with everyday concepts which exhibit graded category membership, prototype effects, and lack authoritative consensus annotations. While our theoretical framework is domain-agnostic, the present application targets a selection of scientific domains relevant to pharmaceutical R\&D (biology, chemistry, and medicine) where ground truth can be reliably established based on expert-curated consensus annotations."
>
> ### Added to manuscript Methods section (lines 144-149):
>
> > "In total 187 concepts were selected from the domains of chemistry, biology and medicine spanning 3 to 151,204 referents per concept (Supplementary Table \ref{table:concepts}). These domains were selected beyond practical relevance to pharmaceutical R\&D: (1) availability of human expert-curated ground truth annotations for concepts (e.g., from ChEBI, MeSH, Gene Ontology), (2) well-defined referents with discrete category membership and consistency between definitions and class members, (3) conceptual diversity across scales (molecular to biological to clinical), and (4) transferability to several key scientific domains (chemistry, biology, medicine)."
>
> ### Added to manuscript Limitations section (line 526-530):
>
> > "The subset of concepts included in the evaluation is small and by no means exhaustive. The main utility of the present work lies in providing a logical framework, proof-of-concept study and a starting point for more comprehensive benchmarks. However, the statistical power of our analyses was very high, as evidenced by extremely small FDR-corrected q-values (e.g., $q < 10^{-14}$ for identifying low-performing models, $q < 0.002$ for identifying high-performing models) and the significance of 26 out of 29 category-level effects after FDR correction, suggesting that increasing the concept selection in the chosen domains would not substantially alter our statistical conclusions."
>
> For overview, below are the key points demonstrating the diversity of the present dataset:
>
> 1. **Substantial diversity of scientific concepts**:
>    - Three distinct knowledge domains (chemistry, biology, medicine)
>    - Wide semantic field size range: 3 to 151,204 referents (5 orders of magnitude)
>    - Multiple conceptual types: molecular entities, biological processes, cellular components, diseases, therapeutic agents
>
> 2. **Authoritative ground truth from expert-curated databases**:
>    - **ChEBI**: 195,000+ manually curated chemical entities (EMBL-EBI expert biocurators)
>    - **MeSH**: 30,000+ terms curated by NLM medical indexers for 60+ years, used to index 35+ million MEDLINE/PubMed citations
>    - **Gene Ontology**: NHGRI-funded consortium with rigorous quality control since 1998
>    - Each annotation represents collective expert consensus (thousands of person-years)
>
> 3. **High statistical power**:
>    - FDR-corrected q-values extremely small (q < 10⁻¹⁴ for low performers, q < 0.002 for high performers)
>    - 26 out of 29 concept category-level effects significant after FDR correction (new analysis on the quantification of the effect of concept categories on performance was added - see below)
>    - Additional concepts in chosen domains would not substantially alter statistical conclusions
>
> 4. **Framework is domain-agnostic by design**:
>    - Applies to any concept with: label, definition, enumerable referents
>    - Tasks systematically derived from formal framework (not domain-specific)
>    - Successfully applied to three distinct technical domains demonstrates transferability
>
> 5. **External validity demonstrated**:
>    - Strong correlations with established benchmarks (τ=0.82 with MMLU, τ=0.39 with GPQA)
>    - Benchmark captures skills related to but distinct from general knowledge
>    - 18% unexplained variance with MMLU indicates measurement of conceptual coherence beyond factual recall

---

> ### Author Response · Authors · 2025-11-25
> **Response to Weakness 3**
>
> Weakness 3: "The paper provides extensive case studies, which are valuable for intuitively illustrating the model outputs. However, I think there needs a more quantitative analysis, such as identifying which types of concepts LLMs perform better on and exploring potential common patterns. The quantitative analysis would make the findings more insightful."
>
> RESPONSE:
>
> We have completed a comprehensive quantitative analysis (new Appendix Section "Quantitative Analysis of Concept Categories").
>
> ### Added to manuscript Results section (lines 264-268):
>
> > "To investigate whether various characteristics of concepts influence task performance, a statistical analysis of category-level effects across five categorization dimensions (abstraction level, semantic field size, domain, ontological structure, and number of selection criteria) was conducted. Overall, abstraction level and semantic field size had the largest and most consistent effects across tasks (detailed analysis in Appendix~\ref{app:concept-categories})."
>
> ### Added to manuscript Appendix (lines 1188-1267):
>
> #### B.6 QUANTITATIVE ANALYSIS OF CONCEPT CATEGORIES
> >To address whether specific concept characteristics systematically influence model performance, we
> analyzed response accuracy across five categorization dimensions of concepts.
> > "**Abstraction level** distinguished between concepts with physical/concrete referents (e.g., molecular structures and drugs), abstract entities (e.g., disease categories and cell types), and mixed referents (implication hierarchies containing both abstract and physical entities e.g., chemical compound ontologies). **Semantic field size** was categorized into small ($|R_C| < 50$), medium ($50 \leq |R_C| \leq 500$), and large ($|R_C| > 500$) tiers based on gold-standard referent counts. **Domain** classified concepts according to their scientific field (biology, chemistry, medicine). **Ontological structure** differentiated between concepts with a flat set of referents (direct children only, depth $\leq$ 1) and hierarchically organized referents (nested subcategories, depth $\geq$ 2). **Count of selection criteria** distinguished between concepts with a few (1-2) selection criteria or more (3-6).
> >
> > For each task, we computed mean response accuracy for each instance (level) of a category and performed statistical comparisons using Mann-Whitney U tests (for binary categories) or Kruskal-Wallis H test (categories with more than two levels). We applied false discovery rate correction (Benjamini-Yekutieli procedure) and calculated effect size ratios as the ratio of best-performing to worst-performing category instance means. Since only 3 out of 29 category effects on task-specific performance were insignificant after FDR correction, we considered effects practically significant when the effect size ratio exceeded 1.5 (indicating $\geq$50\% relative improvement in performance between category levels)."
>
> Content continued in the following comment...

---

> > ### Author Response · Authors · 2025-11-25
> > **Response to Weakness 3 (continued)**
> >
> > >**Abstraction Level:**
> > > "Abstraction level showed the strongest and most consistent effects, with all six tasks exhibiting practically significant differences (effect size ratios 1.55--1.78$\times$, all $q < 10^{-35}$). However, the direction of effect varied systematically by task type. Abstract and mixed concepts led to better concept identification from definition or selection criteria than concrete concepts (decide-concept: 1.61$\times$, decide-concept-from-selection-criteria: 1.62$\times$). This likely reflects that abstract concepts have more linguistically explicit definitions, while concrete molecular structures may require chemical knowledge not captured in text-based definitions. On the other hand, concepts with concrete referents triggered more accurate referent enumeration than those with abstract or mixed referents (limited-list-referents: 1.73$\times$ and limited-list-referents-from-selection-criteria: 1.78$\times$)."
> > >
> > >**Semantic Field Size:**
> > > "Semantic field size exhibited the largest effect size overall with the enumeration of referents for concepts with small semantic fields outperforming large ones. This pattern likely reflects both computational difficulty (retrieving 10 referents from a set of 50 versus 50,000) and knowledge coverage (smaller concepts tend to be more specialized and better documented). For concept identification tasks (decide-concept and decide-concept-from-selection-criteria) the opposite pattern was true indicating that it was easier for the models to identify the label for concepts with large semantic fields based on a definition or selection criteria."
> > >
> > >**Domain:**
> > > "Domain effects were moderate but consistent. Chemistry concepts showed lowest performance on four of six tasks, with particularly strong deficits in referent-related tasks (decide-referents: 1.54$\times$ worse than biology; limited-list-referents-from-selection-criteria: 1.51$\times$ worse than biology; semantic-field-size: 1.58$\times$ worse than medicine). This pattern suggests that chemical nomenclature, structural formulas, and systematic naming conventions present unique challenges not fully captured in text-based training data. It also aligns with anecdotal observations from subject matter experts regarding the knowledge gaps of language models in chemistry relative to biology, for example. Biology concepts generally performed best, likely reflecting both the prevalence of biological discussions in web text and the more linguistically explicit nature of biological definitions compared to chemical nomenclature."
> > >
> > >**Ontological Structure:**
> > > "Ontological structure showed opposing effects depending on task requirements. Concepts with hierarchical referent structures were associated with better concept identification from definitions (decide-concept: 1.60$\times$, decide-concept-from-selection-criteria: 1.46$\times$) while flat structures performed better in referent enumeration tasks (limited-list-referents: 1.43$\times$, limited-list-referents-from-selection-criteria: 1.49$\times$). Hierarchical referent structure might be related to richer semantic context around the concepts in the training data but it is difficult to assess."
> > >
> > >**Selection Criteria:**
> > > "The number of selection criteria (defining features) showed negligible effects across all tasks (effect size ratios 1.01--1.14$\times$, only 2 of 6 tests significant after FDR correction). It suggests that models handle definitions decomposable into 1-6 selection criteria with similar effectiveness implying adequate compositional understanding at the level of conjunctive feature combinations."
> >
> > The updated manuscript includes tables 5 and 6 in the Appendix (lines 1306-1366) showing effect size ratios and concept distribution across categories.

---

> ### Author Response · Authors · 2025-11-26
> **Summary of the updated manuscript**
>
> Dear Reviewer ya6A,
>
> We want to emphasize that we have submitted an **updated manuscript PDF** (dated November 25, 2025) that comprehensively addresses all weaknesses and questions you raised. All additions are highlighted in **blue font** for easy identification.
>
> **Major new contributions added in response to your review:**
>
> 1. **Comprehensive Quantitative Analysis of Concept Categories** (Appendix B.6, lines 1188-1267): In direct response to your Weakness 3 requesting "more quantitative analysis, such as identifying which types of concepts LLMs perform better on," we conducted an extensive systematic analysis examining five categorization dimensions:
>    - Abstraction level (concrete/abstract/mixed)
>    - Semantic field size (small/medium/large)
>    - Domain (biology/chemistry/medicine)
>    - Ontological structure (flat/hierarchical)
>    - Number of selection criteria (few 1-2/ several 3-6)
>
>    We used Mann-Whitney U test and Kruskal-Wallis H tests to estimate the significance of each concept category on performance on each task, applied FDR correction to the p-values and calculated effect size ratios for all 29 category-task combinations. Key findings: chemistry concepts show consistently lower performance (effect size ratios up to 1.58×), abstraction level has the strongest effects (1.55-1.78×), and 26 out of 29 effects remain significant after FDR correction.
>
> 2. **Systematic Investigation of SFS Task Anomalies** (Appendix, lines 1268-1298): We performed statistical testing of multiple hypotheses explaining SFS inconsistencies, including rejection of the gold standard incompleteness hypothesis and identification of domain-dependent instruction ambiguity as a core issue.
>
> 3. **Detailed Comparison with COPEN and OntoProbe** (Discussion, lines 443-453): In response to your Weakness 1, we added comprehensive comparison with the prior works you cited, clearly differentiating our framework's focus on definitional coherence and generative enumeration from their classification-based approaches.
>
> **Additional key revisions:**
> - Domain selection rationale with four specific criteria (lines 144-149)
> - Enhanced scope clarification (Introduction, lines 40-45)
> - Sufficient statistical power note in Limitations (lines 526-530)
> - Comparison with representation geometry work (lines 470-480)
>
> These substantial additions directly address your request for deeper quantitative insights and clearer positioning relative to prior work. We believe the new quantitative analysis significantly strengthens the manuscript's empirical contributions.
>
> Best regards,
> The Authors

---

### Official Review · Reviewer_paTX · 2025-10-31

**Soundness:** 3
**Presentation:** 3
**Contribution:** 3
**Rating:** 4
**Confidence:** 3

**Summary:**

This paper introduces a logical framework of conceptual semantics that formalizes the relationship between concept labels, definitions, and referents. Based on this framework, the authors design a conceptual integrity benchmark comprising six tasks that test semantic mappings between these components. They evaluate 15 LLMs using 187 concepts from biology, chemistry, and medicine, finding that conceptual integrity correlates strongly with model size and with general benchmarks like MMLU. They also identify systematic failure modes (in annotation, scoring, and model reasoning) and discuss implications for ontology completeness and conceptual understanding.

**Strengths:**

* The formalization of conceptual integrity using first-order logic and the link to instruction-tuned benchmarks is original and timely.

* The benchmark spans multiple domains and models, with thoughtful correlation analysis to external metrics (MMLU, GPQA, etc.).

* The discussion on failure modes (especially semantic nuance and ontology incompleteness) is detailed and empirically motivated.

**Weaknesses:**

* Only 187 concepts across three domains, with hand-curated annotations. This limits generalizability.

* Scoring relies on incomplete and potentially inconsistent ontologies, leading to lower-bound estimates.

* Some benchmark tasks (JSON formatting, instruction following) test general compliance rather than conceptual understanding per se.

* The framework defines what to measure but doesn’t explain how models represent or manipulate conceptual structure.

**Questions:**

* Could the benchmark be expanded beyond biomedical/chemical domains to assess transferability?

* How might conceptual integrity relate to known phenomena like compositional generalization or systematicity?

* Could fine-tuning on ontology-style data improve conceptual integrity scores?

* How do annotation or scoring inconsistencies affect the reliability of correlations with other benchmarks?

---

> ### Author Response · Authors · 2025-11-19
> **Response to Weakness 1**
>
> Weakness 1: "Only 187 concepts across three domains, with hand-curated annotations. This limits generalizability."
>
> RESPONSE:
>
> While 187 concepts may appear limited, our benchmark exhibits substantial diversity: three fundamentally different domains (chemistry, biology, medicine), several expert-curated sources (ChEBI: 195K+ entities, MeSH: 30K+ terms, Gene Ontology, cancer.gov), and semantic field sizes spanning 5 orders of magnitude (3 to 151,204 referents). These sources collectively represent hundreds of person-years of expert curation and a gold-standard ground truth unavailable in most knowledge domains.
>
> The framework is domain-agnostic by design, applying to any concept with (1) a linguistic label, (2) definition, and (3) enumerable referents. Transferability to other domains is straightforward when there exists relevant consensus data that provides concept annotations (e.g. legal terms, mathematical concepts, engineering standards). However, for domains lacking expert consensus on definitions and referents, no benchmark approach can provide reliable ground truth and this is a limitation of human knowledge, not our framework.
>
> Finally, we do not consider the usage of hand curated annotations as a serious limitation. As the reviewer Lc1i notes, human baseline is essential for assessing conceptual understanding in non-human systems. It would be rather easy to construct a baseline automatically by asking LMs to provide definitions and referents of concepts. However, this baseline would be biased towards the knowledge represented in the specific LLMs. Of course, by eliciting responses from a group of LMs one would get more a representative baseline but it would still require an impartial expert to resolve the discrepancies between multiple annotations of the same concept. In other words, there seems to be no better way to obtain a gold standard than by using human expert-annotated consensus knowledge.
>
> ### Added to manuscript Limitations section (line 526-530 in updated pdf):
>
> > "The subset of concepts included in the evaluation is small and by no means exhaustive. The main utility of the present work lies in providing a logical framework, proof-of-concept study and a starting point for more comprehensive benchmarks. However, the statistical power of our analyses was very high, as evidenced by extremely small FDR-corrected q-values (e.g., $q < 10^{-14}$ for identifying low-performing models, $q < 0.002$ for identifying high-performing models) and the significance of 26 out of 29 category-level effects after FDR correction, suggesting that increasing the concept selection in the chosen domains would not substantially alter our statistical conclusions."
>
> For overview, below are the key points demonstrating the diversity of the present dataset:
>
> 1. **Substantial diversity of scientific concepts**:
>    - Three distinct knowledge domains (chemistry, biology, medicine)
>    - Wide semantic field size range: 3 to 151,204 referents (5 orders of magnitude)
>    - Multiple conceptual types: molecular entities, biological processes, cellular components, diseases, therapeutic agents
>
> 2. **Authoritative ground truth from expert-curated databases**:
>    - **ChEBI**: 195,000+ manually curated chemical entities (EMBL-EBI expert biocurators)
>    - **MeSH**: 30,000+ terms curated by NLM medical indexers for 60+ years, used to index 35+ million MEDLINE/PubMed citations
>    - **Gene Ontology**: NHGRI-funded consortium with rigorous quality control since 1998
>    - Each annotation represents collective expert consensus (thousands of person-years)
>
> 3. **High statistical power**:
>    - FDR-corrected q-values extremely small (q < 10⁻¹⁴ for low performers, q < 0.002 for high performers)
>    - 26 out of 29 concept category-level effects significant after FDR correction (new analysis on the quantification of the effect of concept categories on performance was added - see below)
>    - Additional concepts in chosen domains would not substantially alter statistical conclusions
>
> 4. **Framework is domain-agnostic by design**:
>    - Applies to any concept with: label, definition, enumerable referents
>    - Tasks systematically derived from formal framework (not domain-specific)
>    - Successfully applied to three distinct technical domains demonstrates transferability
>
> 5. **External validity demonstrated**:
>    - Strong correlations with established benchmarks (τ=0.82 with MMLU, τ=0.39 with GPQA)
>    - Benchmark captures skills related to but distinct from general knowledge
>    - 18% unexplained variance with MMLU indicates measurement of conceptual coherence beyond factual recall

---

> ### Author Response · Authors · 2025-11-19
> **Response to Weakness 2**
>
> Weakness 2: "Scoring relies on incomplete and potentially inconsistent ontologies, leading to lower-bound estimates."
>
> RESPONSE:
>
> We acknowledge this important limitation and have addressed it with both conceptual clarification and an in-depth analysis of semantic-field size task anomalies that we have added in response to multiple requests from reviewers (lines 1268-1298 in the updated pdf).
>
> ### Added to manuscript Appendix (lines 1280-1289 in the updated pdf, all additions in blue font):
>
> > "First, we tested the hypothesis that SFS task performance inconsistency might be due to the incompleteness of the expert-curated gold standard. As exhaustive enumeration of referents is much harder for concepts with thousands rather than dozens of referents, one would expect to see higher normalized discrepancy for concepts with larger semantic fields. To that end, we partitioned concepts by semantic field size into small ($|R_C|$ < 50), medium (50 <= $|R_C|$ <= 500) and large semantic field groups ($|R_C|$ > 500). Mann-Whitney U test of whether the discrepancy was higher in large vs small semantic fields was highly insignificant (p = 1.000). Spearman correlation between semantic field size and normalized discrepancy was significantly negative ($\rho$ = -0.178, p = 6.105e-21), indicating that point estimates for concepts with larger semantic fields exhibited marginally better agreement with gold standard. These results contradict the incompleteness hypothesis, suggesting that gold standard quality is unlikely to be the primary issue behind inconsistencies."
>
> **Key points from the manuscript addressing this concern:**
>
> 1. **Empirical evidence against systematic bias**: Statistical testing rejected the hypothesis that large semantic fields (where incompleteness would be most problematic) show disproportionate performance issues. The weak negative correlation (ρ = -0.178) supports this observation.
>
> 2. **Conservative estimates affect all models equally**: While ontologies provide lower-bound performance estimates (false negatives when models produce correct referents not in databases), this limitation affects all evaluated models equally, preserving relative rankings.
>
> 3. **State-of-the-art gold standards**: The ontologies used (ChEBI, MeSH, Gene Ontology) represent expert-curated consensus maintained by domain specialists, representing the best available ground truth.
>
> 4. **LLM-as-judge scoring mitigates literal matching issues**: Our scoring approach accepts semantically equivalent paraphrases beyond literal database matches, reducing inconsistencies from minor variations.
>
> 5. **Fundamental limitation of knowledge representation**: Any ontology summarizing domains with potentially unlimited referents is necessarily incomplete. Perfect completeness is impossible—alternatives such as crowdsourced or automatically generated annotations would introduce even greater inconsistencies.

---

> ### Author Response · Authors · 2025-11-19
> **Response to Weakness 3**
>
> Weakness 3: "Some benchmark tasks (JSON formatting, instruction following) test general compliance rather than conceptual understanding per se."
>
> RESPONSE:
>
> This is accurate: our benchmark tests JSON formatting and instruction following alongside conceptual integrity. However, for practical deployment in knowledge work, this is a feature, not a bug. Real-world applications require the complete pipeline: understanding concepts, following instructions, and producing structured outputs. For pharmaceutical R&D (our motivating application), these "accessory skills" are essential. Our benchmark identifies deployment-ready models, not just ontology-focused models specializing in conceptual knowledge.
>
> Alternative task designs could address conceptual knowledge in less demanding skill settings (e.g., multiple-choice answers) but they would sacrifice assessment of generative capabilities and practical task requirements. In our opinion, capacity for structured output and instruction following are basic skills that any model seriously contending to automate knowledge work needs to bring to the table.

---

> ### Author Response · Authors · 2025-11-19
> **Response to Weakness 4**
>
> Weakness 4: "The framework defines what to measure but doesn't explain how models represent or manipulate conceptual structure."
>
> RESPONSE:
>
> Indeed, our framework is definitional and evaluative, not mechanistic. This is intentional since rigorous definition and measurement lays the groundwork to any attempts of mechanistic investigation.
>
> Our contribution provides:
> - Formal theory of conceptual semantics grounded in first-order logic with machine-verified proofs
> - Systematic method for deriving benchmark tasks from theoretical principles
> - Empirical measurements enabling model selection and progress tracking indicating statistically significant performance differences between models and reasonable agreement with established benchmarks
> - Foundation for future mechanistic studies (probing, interventions, training dynamics etc.)
>
> The framework enables but does not conduct mechanistic investigation which is valuable future work building on this foundation. We believe establishing rigorous measurement methods for conceptual integrity will enable the mechanistic interpretability community to investigate how models actually represent and manipulate conceptual structures. For example, we would love to see studies that attempt to determine how are the semantic field size estimates represented on the level of contextualized embeddings and compare this between sibling models of different sizes. Without the proposed logical framework, such problems would be considered dubious or, in worst case, even meaningless.

---

> ### Author Response · Authors · 2025-11-19
> **Response to Question 1**
>
> Question 1: "Could the benchmark be expanded beyond biomedical/chemical domains to assess transferability?"
>
> RESPONSE:
>
> Yes, the framework is theoretically domain-agnostic as it applies to any concept with (1) a linguistic label, (2) selection criteria (definition), and (3) enumerable referents. Our three-domain demonstration (chemistry, biology, medicine) shows the framework can be applied across different knowledge domains with physical, clearly decidable referents. It would be straightforward to extend the benchmark to domains with similar characteristics (e.g., with physical referents, expert consensus on definitions, authoritative enumeration).
>
>  However, we do acknowledge potential limitations:
>
> - **Everyday concepts** (e.g., "furniture," "game") pose challenges due to graded category membership, prototype effects, and lack of expert consensus on definitional boundaries. Cases where standard usage is not based on definition but arbitrary set membership are more related to factual knowledge representation than representing a consistent ontology.
>
> Nevertheless, we can also imagine how the proposed benchmark can be at least partially adapted to the benchmarking of LLMs with respect to everyday concepts that are vaguely defined although this was not the intention of the present study. As the reviewer Lc1i notes, cognitive psychologists have been collecting semantic feature norms (human subjects' opinions about the properties of various concepts) to characterize the semantic constituents of everyday concepts. Each observed feature norm is represented by a frequency among the sampled subjects. We can treat the frequency of each norm as a score from 0 to 1 awarded to the model if it is able to reproduce this norm when asked to think what properties are important for the given concept (the same instruction is given to the human subjects). If we award the corresponding frequency to the LLM subject as a score for each unique response that maps to the feature norms from the human sample then the average of the norm scores for a single concept would approximate an accuracy measure of how well the LLM responses align with the distribution of feature norms from the human subject. This approach would enable the benchmarking of feature norms similarly to what we have done but it would not map neatly to the underlying logical formalism because feature norms are not logically equivalent to selection criteria as defined in the present framework (please see response titled "Human grounding of concept annotations" to reviewer Lc1i). However, the latter is not a weakness of the present approach, rather a weakness of the lack of logical consistency of the feature norm approach.
>
> - **Abstract concepts** (e.g., mathematical objects, social constructs, philosophical ideas) may lack the physical, easily decidable referents that characterize our current domains. For example, referents of the concept of circle defined mathematically with absolute precision might not be found in the physical world. On the other hand, we can have an infinite number of distinct equations satisfying the general equation of the circle due to an infinite number of values for the radius.
>
> - **Emerging or contested domains** may lack mature ontologies or standardized terminology
>
> The critical constraint is availability of authoritative ground truth: without a consensus on definitions and referents, any benchmark becomes dependent on arbitrary annotation choices. This is a limitation not of our framework specifically, but of evaluating conceptual understanding in domains where human knowledge itself lacks consensus.

---

> ### Author Response · Authors · 2025-11-19
> **Response to Question 2**
>
> Question 2: "How might conceptual integrity relate to known phenomena like compositional generalization or systematicity?"
>
> RESPONSE:
>
> Compositional generalization refers to the ability to understand and produce novel combinations of familiar components by systematically combining known parts according to learned rules [1]. It rests on the systematicity argument of Fodor and Pylyshyn [2] who proposed that systematicity arises from the compositional structure of mental representations. When thoughts are built from reusable constituents in systematic ways (provided some semantic constraints), having the ability to form one thought guarantees the ability to form related thoughts with the same constituents in different arrangements.
>
> Conceptual integrity framework as proposed here deals explicitly with the compositionality of meaning on the concept level by representing its definition as a set of decidable selection criteria for the referents. By adding or removing selection criteria, more or less specific concepts are created resulting in a conceptual hierarchy. Similarly to the examples presented in [2] (e.g. "red square"), conceptual integrity extends compositionality to noun phrase level by treating noun modifiers as additional selection criteria piled on top of those from the core noun to generate increasingly specified concepts. Similarly, adverbs and prepositional phrases specifying time and place are used to modify verbs to specify predicates.
>
> In summary, conceptual integrity enables recursive definitions of concepts in terms of other concepts all the way down to the most elementary concepts (called [qualia](https://plato.stanford.edu/entries/qualia/) by some philosophers). The selection criteria of these elementary concepts are rooted in perception (e.g. the only selection criterion for the color blue would be "it is blue") serving as the base case and preventing infinite regress. In everyday life, one can often come across concepts that seem to be loosely defined in terms of "[I know it when I see it](https://en.wikipedia.org/wiki/I_know_it_when_I_see_it)" supporting the idea of the elementary selection criteria being rooted in perception.
>
> We have added the following paragraph to the discussion (lines 502-516 in the updated pdf):
>
> >"On a higher level, an apparent connection exists between conceptual integrity and compositional
> generalization which is the ability to understand novel combinations of known components system-
> atically combined according to learned rules Lake & Baroni (2018). Similarly to the systematicity
> argument of Fodor and Pylyshyn Fodor & Pylyshyn (1988), our framework treats conceptual integrity
> as explicitly compositional: definitions are sets of decidable selection criteria, and adding or removing
> criteria creates more or less specific concepts in a hierarchy. This extends naturally to noun phrase
> compositionality where noun modifiers recruit additional selection criteria above and beyond those of
> the core noun. Crucially, this compositional structure must ultimately ground in elementary concepts
> to avoid infinite regress. We propose that these elementary concepts what some philosophers call
> qualia Tye (2021) have selection criteria rooted directly in perception (e.g., the concept ‘blue’ is
> defined simply as ‘it is blue’). This grounding in perception provides the base case for recursively
> defined concepts and aligns with everyday usage where vague concepts are defined as ‘I know it
> when I see it’ Wikipedia contributors (2024). Our framework thus positions conceptual integrity as
> a foundational capability underlying both compositional generalization and systematic reasoning:
> accurate understanding of constituent concepts (including perceptually grounded elementary ones) is
> prerequisite to valid reasoning about their combinations."
>
>
> **References:**
> - [1] Lake, B. M., & Baroni, M. (2018). Generalization without systematicity: On the compositional skills of sequence-to-sequence recurrent networks. ICML. https://proceedings.mlr.press/v80/lake18a.html
> - [2] Fodor, J. A., & Pylyshyn, Z. W. (1988). Connectionism and cognitive architecture: A critical analysis. Cognition, 28(1-2), 3-71. https://doi.org/10.1016/0010-0277(88)90031-5

---

> ### Author Response · Authors · 2025-11-19
> **Response to Question 3**
>
> Question 3: "Could fine-tuning on ontology-style data improve conceptual integrity scores?"
>
> RESPONSE:
>
> Very likely. Prior work on ontological knowledge probing provides strong evidence for the benefits of training on structured data:
>
> **COPEN** [1] evaluated PLMs using three probing methods:
> 1. Zero-shot probing: tests intrinsic knowledge without training (lower bound)
> 2. Linear probing: trains only a classification head on top of frozen PLM representations
> 3. Fine-tuning: Trains all PLM parameters on the training data (upper bound)
>
> Their key finding: Fine-tuning performance substantially exceeded zero-shot and linear probing, demonstrating that "PLMs will inevitably fit the probing tasks through the information in training data rather than only resort to their intrinsic knowledge." This improvement held across model scales, though even the largest models (T5-11B with 11 billion parameters) remained well below human performance after fine-tuning.
>
> **OntoProbe** [2] similarly used training data for soft prompt tuning, finding that models can improve their ontological reasoning when exposed to structured examples during training. Their work on memorization vs. reasoning showed that explicit training on ontological structures improves both recall and logical inference capabilities.
>
> **Implications for our work:**
>
> Fine-tuning on ontology-structured data would likely improve scores substantially but with some caveats:
>
> 1. it is unknown to what extent the training data of SOTA models already includes ontological relationships from the knowledge bases used in the present study. Presumably, it makes a difference whether the model has to infer conceptual relationships from the contextualized token representations or they are "hard-coded" (memorized by concept-specific attention heads) by overfitting to the training data.
>
> 2. Fine-tuning of models on ontologies might improve conceptual integrity performance (by helping them memorize relevant structures) while decreasing performance in other areas (e.g., reasoning) and without leading to compositional generalization.
>
> 3. On the other hand, if the contextualized representations of tokens were to encode semantic cues in a deeply compositional manner, one would expect that encouraging models via fine-tuning to use this information systematically (e.g., to estimate conceptual similarity by estimating the overlap between the referents of two concepts) might lead to superior conceptual similarity judgements.
>
> **References:**
> - [1] Peng, H., et al. (2022). COPEN: Probing conceptual knowledge in pre-trained language models. EMNLP. https://aclanthology.org/2022.emnlp-main.335/
> - [2] Wu, W., et al. (2023). Do PLMs know and understand ontological knowledge? ACL. https://doi.org/10.18653/v1/2023.acl-long.173

---

> ### Author Response · Authors · 2025-11-19
> **Response to Question 4**
>
> Question 4: "How do annotation or scoring inconsistencies affect the reliability of correlations with other benchmarks?"
>
> RESPONSE:
>
> **Annotation inconsistencies**
>
> Annotation inconsistencies primarily cause false negatives (lowering absolute scores while preserving relative rankings) leading the benchmark to report a lower bound of performance. Our mitigation strategies include:
>
> 1. multiple knowledge domains (biology, chemistry, medicine) and multiple expert sources - annotation errors are expected to be independent between domains and expert sources leading to reduced systematic bias
>
> 2. LLM-as-judge to establish semantic equivalence between response and gold standard: more robust than symbolic/keyword-based evaluation of equivalence
>
> 3. limited enumeration tasks letting LMs respond with entities that are of highest probability (most frequent in the training corpus) and preventing them from descending into repetitive loops. It effectively makes the task much easier when compared to exhaustive enumeration, because the models can respond with "usual suspects" instead of having to come up with rare/exotic instances.
>
> **Scoring inconsistencies**
>
> Scoring inconsistencies introduce measurement noise across the board and are expected to reduce correlations with external benchmarks as well as the internal consistency (correlations between performance on different knowledge domains addressed in the benchmark).
>
> The strong internal and external correlations (τ=0.82 with MMLU, τ=0.39 with GPQA) we observe despite imperfect measurement indicate:
>
> 1. sufficient validity with reference to external benchmarks
> 2. internal consistency based on high performance correlation between different knowledge domains
> 3. robust model rankings (Kendall's τ is more resistant to noise than parametric correlation coefficients)

---

> ### Author Response · Authors · 2025-11-26
> **Summary of the updated manuscript**
>
> Dear Reviewer paTX,
>
> We want to ensure you are aware that we have submitted an **updated manuscript PDF** (dated November 25, 2025) that addresses all points raised in your review. All additions are highlighted in **blue font** throughout the manuscript for easy identification.
>
> **Major new analyses added:**
>
> 1. **Comprehensive Quantitative Analysis of Concept Categories** (Appendix B.6, lines 1188-1267): We conducted a systematic statistical analysis examining how five concept characteristics (abstraction level, semantic field size, domain, ontological structure, and number of selection criteria) influence task performance across all models. This analysis includes effect size calculations and FDR-corrected significance testing, revealing that abstraction level and semantic field size have the largest and most consistent effects across tasks (effect size ratios 1.43-1.78).
>
> 2. **Systematic Investigation of Semantic Field Size Task Anomalies** (Appendix, lines 1268-1298): We performed an in-depth analysis testing multiple hypotheses about SFS task inconsistencies, including statistical tests rejecting the gold standard incompleteness hypothesis and identifying systematic instruction ambiguity arising from fundamental differences in ontological conventions across domains (class enumeration vs. physical instance counting).
>
> **Key manuscript revisions:**
> - Enhanced Limitations section with note about statistical power (lines 526-530)
> - Added discussion of compositional generalization and systematicity (lines 502-516)
> - Expanded Discussion with comparison to COPEN and OntoProbe frameworks (lines 443-453)
> - Multiple clarifications throughout addressing raised weaknesses and questions
>
> We believe these substantial improvements significantly strengthen the manuscript's empirical foundation and address the concerns about limited scope and quantitative depth. We hope you will consider these enhancements when finalizing your evaluation.
>
> Best regards,
> The Authors

---

### Meta-Review · Area_Chair_2Hxk · 2025-12-15

**Summary:**

The paper proposes to probe the boumdaries of concepts in language models and provides a framework with first -order logics. There are 4 reviewers that give feedbacks on the paper. Three of four hold negative score and one hold positive scores. There are some major concerns which are important.
1) Limitation of generalizability:  Reviewer paTX and ya6A show that only using 187 concepts may affect the generalizability of the experimental results.
2) Lack of reliable ground truth to evaluate model responses:  Reviewer  Lc1i has raised this concern while a suitable ground truth should be placed for fair evaluation
3) Scoring inconsisrences: Reviewer paTX comments that annotation or scoring inconsistencies may affect the results of the framework
4) Deep analysis with other  benchmarks such as MMLU or GPQA(Reviewer gJfc)

While the authors have endeavored to address the concerns raised by the reviewers, several key limitations identified in the review process remain unresolved. These include issues pertaining to the generalizability, the absence of reliable ground truth data, and inconsistencies in scoring metrics. Consequently, the paper is rejected.

**Reviewer Concerns:**

The following  concerns are addressed:
1) Reviewer paTX: paTX raises 4 weaknesses and 4 questions. To the best of my efforts,  some of the weakness such as W2,W4 are addressed while W1 and W3 are somewhat unsatisfying.  I think W1 is outstanding which should be carefully considered.
2) Reviewer ya6A: ya6A raises 3 weaknesses. To the best of my knowledge, the reviewer may focuse the experimental difference between the proposed framework and recent works thus to show the contributions.  Also more quantitative analysis should be conducted. For the rebuttals, W1 is partially addressed,  some experimental results can be more supportive. For W2,  though the authors argue that the iuncluded concepts come from a wide range scopes, it can not resolve the issue the limited data size. For W 3,  the reviewer may want to see some table like results but the author may provide with textual results.  For Weakness 2, I think it is outstanding, data size may also be important;
3) Reviewer Lc1i: Lc1i raised 3 weaknesses and 5 questions. For weaknesses , I think,  the author does not give a valid point-to-point feedbacks. Although the Reviewer has aggreed to improve the score. But the reviewer still have concerns about the generality.
4) Reviewer  gJfc: gJfc raises 2 questions and some weaknesses. The reviewer showed positive feedbacks but keep the origin score.

**Reviewer Scores:**

After reading the rebuttals from the reviewers, I think, none of  them would improve their score and keep their origin scores as is. Although the author spends much space trying to answer the questions and weaknesses, the issues still remain such as the generability,Lack of reliable ground truth  and Scoring inconsisrences, which I think, may be the key factors that affecting the reviewers' score. Indeed, only Reviewer gJfc shows a positive feedback but  he does not improve his score.  And Reviewer Lc1i is willing to improve the score but still have  concerns about the generalizability of this approach and whether this is even the right kind of framework to be pursuing.  For other reviewers,  it is unlike to improve the scores.

---

### Decision · Program_Chairs · 2026-01-26

Reject